# RAPTORGRAPH: GRAPH-BASED PATHWAY MODELING FOR CAUSAL DISCOVERY IN SINGLE-CELL PERTURBATIONS

## ABSTRACT

Experiments involving the perturbation of individual cells are central to understanding cellular mechanisms and can accelerate drug discovery. Causal representation learning (CRL) allows us to uncover the latent factors that regulate biological systems and predict the impact of novel perturbations. Unfortunately, existing methods fail to address intervention spillover in a closed-world setting where intervention targets are known *a priori*, such as in Perturb-seq experiments, due to their reliance on dense encoders. Furthermore, incorporating curated biological pathways into the model imposes a confirmatory bias, forcing it to explain the data through preexisting pathways and reducing the set of hypotheses the model can explore, while discarding novel signals that lie outside the annotated pathways. In this work, we introduce RAPTORGraph, a $\beta$-VAE with a GraphPathway encoder that explicitly models complex gene-to-gene interactions within learned pathways. Moreover, our model's preconditioning isolates the influence of perturbed genes, yielding clean, single-node latent interventions required for identifiable causal discovery and eliminating spillover. Finally, we train the model on data preprocessed with optimal-transport alignment, which guarantees a well-defined mapping between control and perturbed samples and further stabilizes the learned latent representations. We demonstrate that RAPTORGraph improves state-of-the-art performance on downstream analyses of unseen perturbations, such as non-additive interactions, while outperforming other approaches on objective metrics, such as MSE and MK-MMD. The code will be made publicly available upon publication of this paper.

## 1 INTRODUCTION

Understanding how cells respond to genetic perturbations, such as those induced by CRISPR screens (Dixit et al., 2016), is fundamental to advancing our understanding of basic biology and therapeutic discovery (Plenge et al., 2013; Adamson et al., 2016; Norman et al., 2019). However, the combinatorial explosion across ∼20,000 protein-coding genes makes exhaustive perturbation experiments infeasible (Norman et al., 2019), motivating computational models that predict cellular responses to unseen perturbations or combinations thereof.

Existing approaches to predict the transcriptional outcome following a perturbation face a fundamental challenge: generative quality comes at the cost of causal interpretability. For example, recent methods like GEARS (Roohani et al., 2022) or scGPT (Cui et al., 2024) achieve strong predictive performance in cellular response to perturbations; however, they act as "black boxes" providing little mechanistic insight into how perturbations affect cellular systems. This opacity prevents researchers from understanding why certain combinations of perturbations produce synergistic effects, failing to provide testable biological hypotheses.

As such, computational biology demands models that provide mechanistic insights into cellular function (Chen et al., 2024), making causal representation learning (CRL) (Schölkopf et al., 2021) a promising approach for mapping genes to biological processes. In particular, recent CRL approaches such as DiscrepancyVAE (dVAE) (Zhang et al., 2023) or SENA (de la Fuente et al., 2025) aim at recovering the underlying causal generative factors and their interactions, enabling counterfactual

reasoning. However, these CRL approaches suffer from fundamental architectural limitations that prevent them from fulfilling their theoretical causal identifiability guarantees: the use of a dense interventional encoder inherently means that every gene, including those intervened upon can influence all latent variables. We term this phenomenon *intervention spillover*, as it renders the task of identifying a unique latent target for an intervention ill-posed. Consequently, any mechanism designed to apply the intervention is forced to converge to a suboptimal solution, because it cannot perform the clean, single-node manipulation required for valid causal inference (Brehmer et al., 2022; Lippe et al., 2022).

Further, the destructive nature of single-cell RNA sequencing (scRNA-seq) makes it impossible to observe the same cell before and after a genetic intervention, complicating the prediction of cell-specific responses. Existing methods address this issue by randomly pairing control and perturbed cells. Since the model must minimize error across all possible arbitrary pairings, it is incentivized to learn the population-mean response. We refer to this phenomenon as *mean collapse*, as this regression to the mean erodes cellular diversity.

We present **R**esponse **A**nalysis of **P**erturbed **T**ranscript**O**mes using Inte**R**pretable **Graph** (RAPTOR-Graph), a novel framework that addresses causal identifiability challenges in perturbation prediction through structured encoder design. Our first key innovation is the GraphPathway layer, a novel architecture that uses preconditioning to enforce sparse mappings from genes to learned latent factors. This design enables the model to perform the clean, single-node manipulations required for valid causal inference, directly maintaining interpretability. This innovation is complemented by a novel optimal transport (OT) preprocessing step, which mitigates the distinct problem of mean collapse. We benchmark the generative quality of RAPTORGraph on the Norman et al. (2019) Perturb-seq dataset and the Replogle et al. (2020) CRISPRi dataset. For the Norman et al. (2019) data, we further evaluate the accuracy of genetic interaction in perturbation combinations, and the biological interpretability of the inferred latent factor. Specifically, when applied to scRNA-seq data from a K562 cell line, RAPTORGraph uncovered distinct, biologically accurate programs, such as the differentiation-induced cell cycle arrest driven by EGR1 and stress-induced growth arrest mediated by JUN. Additionally, we Replogle et al. (2020) CRISPRi dataset We demonstrate strong performance alongside interpretability, which addresses key gaps not filled by current approaches.

## 2 BACKGROUND

### 2.1 IN SILICO HIGH-THROUGHPUT SCREENING MODELS

Accurate *in silico* prediction of perturbation effects remains critical to guide experimental perturbations assays, as they would allow us to predict, for instance, how cells respond to environmental stress (Frangieh et al., 2021). However, a key limitation in learning perturbation responses lies in the destructive nature of single-cell transcriptomic assays, which prevents measuring the same cell before and after a perturbation. While paired measurements are infeasible, we do have access to independent single-cell observations from control and perturbed cells.

Recent works have demonstrated the potential of representation learning in leveraging these perturbation data to predict transcriptional outcomes of novel combinations of perturbations. For example, Kamimoto et al. (2023) relied on inferring (linear) transcriptional relationships between genes in the form of a gene regulatory network to infer the effect of unseen perturbations. Shortly after, the Compositional Perturbation Autoencoder (CPA) (Lotfollahi et al., 2023b) predicted combinations of perturbations by vector arithmetic in the latent space generated by a Variational Autoencoder (VAE). In parallel, GEARS (Roohani et al., 2024) leveraged Graph Neural Networks (GNNs) to incorporate prior knowledge of gene-gene relationships into the model architecture, thereby predicting unseen genetic interventions and their combinations. More recently, CellOT (Bunne et al., 2023) proposed to directly learn optimal transport maps between control and perturbed cell states, thus accounting for heterogeneous subpopulation structures in the data. Additional efforts, such as scGen (Lotfollahi et al., 2019), scGPT (Cui et al., 2024), have explored Transformer-based approaches for this task.

Even though all the outlined approaches tackle the common problem of predicting the transcriptional consequences of perturbations, their non-causal "black box" nature hampers their capability for uncovering the causal mechanisms driving observed transcriptional changes. This poses a significant limitation in, for example, cellular engineering, where understanding the *why* behind perturbation

effects is as critical as predicting the *what*. To address this challenge, recently, dVAE (Zhang et al., 2023) learns causal latent factors without an explicit semantic meaning, resulting in causally valid but biologically uninterpretable latent representations. SENA (de la Fuente et al., 2025), on the other hand, explicitly models (a set of) biological processes as latent causal variables, promising both predictive accuracy and mechanistic interpretability.

## 2.2 STRUCTURAL CAUSAL MODELS FOR LATENT DISCOVERY

We model the underlying data-generating process using a structural causal model (SCM) (Pearl, 2009). We assume that observed high-dimensional samples $\mathbf{x} \in \mathbb{R}^n$ (e.g., gene expression) are generated by a set of unobserved, low-dimensional latent causal variables $\mathbf{u} \in \mathbb{R}^d$, where $d \ll n$. The SCM defines the relationships between these variables via a set of structural assignments:

$$\mathbf{u}_i = f_i(\mathrm{Pa}_{\mathcal{G}}(\mathbf{u}_i), \mathbf{z}_i), \quad \text{for } i = 1, \ldots, d \tag{1}$$

where $f_i$ is the structural function, $\mathrm{Pa}_{\mathcal{G}}(\mathbf{u}_i)$ are its direct causal parents in a graph $\mathcal{G}$, and $\mathbf{z}_i$ is an independent exogenous noise term. The complete set of these equations induces a directed acyclic graph (DAG) $\mathcal{G}$, which implies that the joint probability distribution over $\mathbf{u}$ factorizes as:

$$\mathrm{p}(\mathbf{u}) = \prod_{i=1}^{d} \mathrm{p}(\mathbf{u}_i \mid \mathrm{Pa}_{\mathcal{G}}(\mathbf{u}_i)) \tag{2}$$

However, these latent causal variables are not directly accessible. Instead, we only observe the high-dimensional data $\mathbf{x}$, which are generated via a true, unknown mixing function $g^*$:

$$\mathbf{x} = g^*(\mathbf{u}) \tag{3}$$

## 2.3 IDENTIFIABILITY FROM TARGETED INTERVENTIONS

The central challenge of CRL is *identifiability*: learning an encoder function $f_\theta$ that can uniquely recover the true latents $\mathbf{u}$ from only observations $\mathbf{x}$. However, using observational data alone, this goal is theoretically intractable (Pearl, 2009). In this sense, Khemakhem et al. (2020) showed that different parameter configurations of a VAE can generate equivalent marginal distributions $\mathrm{p}(\mathbf{x})$, leaving the learned latent representations non-unique. In the causal setting, this problem is compounded, as multiple distinct causal structures can generate identical observational distributions. This means that learning interpretable causal mechanisms from observational data alone is ill-posed.

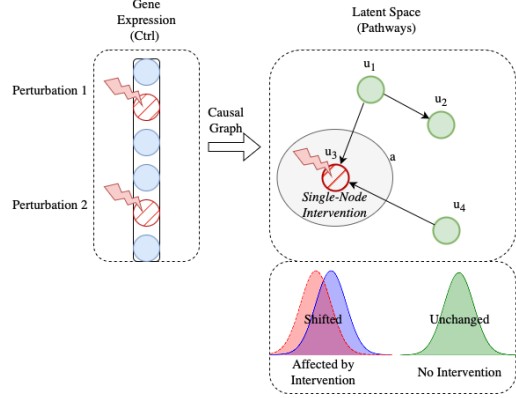

Figure 1: Illustration of an atomic intervention in the latent space. An intervention in the observed gene expression space $\mathbf{x}$ causes a shift in a single latent pathway activity $\mathbf{u}_i$. This alters the conditional distribution $p^I(\mathbf{u}_i \mid \mathrm{Pa}_{\mathcal{G}}(\mathbf{u}_i))$ while leaving unrelated latent variables unaffected, a key requirement for identifiability.

To overcome this fundamental limitation, modern CRL relies on access to interventional data. The key insight is that interventions create unique statistical signatures that break the symmetries present in observational data. As formally established in recent work Schölkopf et al. (2021), achieving identifiability depends on a set of key requirements (see 1).

**Requirement 1 (Acyclic Causal Structure).**
The assumption that the causal structure forms a DAG is a foundational principle in causal discovery (Pearl, 2009).

**Requirement 2 (Atomic Interventions).** Identifiability theory demonstrates that causal structure recovery requires access to interventional data that targets each individual latent variable at least once. This requirement of atomic interventions is a key assumption in modern CRL methods (Zhang et al., 2023; von Kügelgen et al., 2023).

**Requirement 3 (Faithfulness).** The framework also relies on the *faithfulness assumption*, which posits that all conditional independencies in the data are a consequence of the causal graph structure (d-separation) and not due to coincidental cancellations of parameters. The work of Zhang et al. (2023) formalizes two key aspects of this requirement for their setting. First, *linear interventional faithfulness* ensures that the effect of an intervention on a descendant is not perfectly cancelled by a linear combination of other variables. Second, they assume no perfect cancellation between a direct causal effect and other parallel paths in the graph.

Under these conditions, it is possible to provably recover the true latent causal graph up to a well-defined equivalence class. Following the work of Zhang et al. (2023), their main theorem provides a formal guarantee for identifiability:

**Theorem 2.1** (Full DAG Identifiability, Zhang et al. (2023)). *Assume an SCM with a latent DAG $\mathcal{G}$ and an invertible mixing function. If the set of interventions is atomic (targeting each latent variable at least once), and the assumptions of faithfulness hold, then the latent DAG $\mathcal{G}$ and the intervention assignments are identifiable up to permutation, scaling, and translation (the CD-equivalence class).*

This reliance on an **atomic set of interventions**, where each latent variable is targeted at least once, is a cornerstone of modern CRL. It forms the basis for models applied to Perturb-seq data, as seen in recent methods (Zhang et al., 2023; de la Fuente et al., 2025).

## 2.4 THE INTERVENTION SPILLOVER PROBLEM

While identifiability theory provides a strong theoretical foundation, a key practical challenge arises from the architectural choices in most CRL models. We term this the *intervention spillover problem*: the phenomenon where a targeted, single-variable intervention in the high-dimensional observation space ($\mathbf{x}$) is transformed into a dense, multi-node signal in the lower-dimensional latent space ($\mathbf{u}$). For a more detailed theoretical analysis of this problem, see Sec. B.1.

This issue stems from a direct conflict with the atomic intervention requirement discussed in Sec. 2.3. An *Ideal Causal Encoder*, $f^*$, must satisfy this requirement by ensuring that a targeted intervention on a single gene isolates its distributional effect to the parameters of a corresponding single latent variable. Consequently, the expected change in the latent space would be perfectly sparse, manifesting as a clean, single-node perturbation rather than a dense signal. For a rigorous mathematical formulation of this ideal behavior, the formal assumptions, and the proof of incompatibility for dense architectures, we refer the reader to Sec. B.1.1, Sec. B.1.2, and Sec. B.1.3.

**Intervention Spillover Problem in current models**. However, current interventional encoder networks $f_\theta$ are typically dense architectures like MLPs. These encoders possess a dense Jacobian matrix, meaning each component of the latent vector is a function of all input components. Consequently, a sparse change in the input $\mathbf{x}$ propagates through the network, and the model is thus forced to choose the 'best' single target from a dense signal that suggests multiple viable targets.

This architectural mismatch has several critical implications for the reliability of learned intervention targets, which are discussed in detail in Sec. B.1.4. First, models that use a softmax function for target identification are forced into a "winner-take-all" decision on a dense signal, creating a risk of misattributing the primary causal effect. Furthermore, while the perturbed gene is known *a priori*, its corresponding target in the latent space is not, making the task of reliably discovering it from an entangled signal intractable. This reframes the primary modeling objective: the model should be designed not to explore for a latent target, but to accurately learn the magnitude of the intervention's effect on the causal representation. We provided an in-depth discussion about this problem in Sec. B.1.

**Spillover problem in genetic perturbation prediction**. This phenomenon is particularly concerning in genetic perturbation experiments, as interventions are applied to individual genes in the high-dimensional expression space, yet their effects propagate through the cell's regulatory network and manifest as coordinated changes across multiple (latent) biological processes. It is thus crucial to distinguish this intervention spillover, an artifact of dense encoders, from true biological correlation. High correlations between genes do not violate the single-node intervention requirement. Rather, dense encoders fail by transforming a sparse perturbation into a dense entry point in the latent space. By enforcing a sparse mapping for the intervention target, our propose architecture satisfies identifiability while still modeling complex gene-gene correlations downstream via the dense components of the encoder and the learned causal graph.

## 2.5 The Dilemma of Prior Knowledge in Causal Discovery

Existing causal discovery models can be positioned on a spectrum (Herrmann et al., 2024) from purely exploratory approaches like dVAE, which learns all components from data, to more confirmatory models like SENA, which leverages predefined biological pathways to enhance interpretability. While this confirmatory strategy aims to ground the model in known biology, it introduces a significant limitation stemming from poor knowledge transfer. Biological pathways are known to exhibit strong cell-type and cell-line specificity (Schneider et al., 2017; Gamazon et al., 2018; Walter, 2019; Hekselman & Yeger-Lotem, 2020), meaning that mechanisms characterized in one cellular context often do not generalize to another.

## 3 Methods

The RAPTORGraph is an end-to-end framework designed to learn causal relationships from interventional single-cell data. It consists of two novel modules: a preconditioned GraphPathway Encoder (Sec. 3.1 ) that learns a sparse mapping from genes to a set of learned interpretable latent factors (denoted hereafter as meta-pathways) and a DAGMA-based DAG module (Sec. 3.2) that infers the causal graph between the learnt meta-pathways. This design directly addresses the Intervention Spillover Problem ensuring, by construction, that interventions on genes target a unique meta-pathway latent causal factor. We refer to Sec. E.1 for a detailed description and schematic of the framework.

### 3.1 GraphPathway Encoder for Deconfounding through Preconditioning

To address the Intervention Spillover Problem and enforce the theoretical requirement for single-causal-factor interventions, we introduce a novel encoder architecture, the GraphPathway Encoder, which features a deconfounding approach through preconditioning. This approach requires a specific ordering of the input data. Let $k$ be the number of known perturbed genes in the experiment. Without loss of generality, we assume the input gene vector $\mathbf{x} \in \mathbb{R}^n$ is sorted such that the first $k$ genes are those perturbed, followed by the remaining $n - k$ unperturbed genes.

**Pre-conditioning module (Fig. 2-A).** Given this sorted input, we define our model with $d$ learned causal latents (meta-pathways), where we require that the number of learned meta-pathways is at least the number of perturbed genes ($d \geq k$). We then structure the encoder's primary weight matrix $\boldsymbol{W}$ as a block matrix:

$$\boldsymbol{W} = \begin{pmatrix} \Lambda_{11} & \boldsymbol{W}_{12} \\ \mathbf{0}_{(d-k) \times k} & \boldsymbol{W}_{22} \end{pmatrix} \in \mathbb{R}^{d \times n}, \quad \Lambda_{11} = \begin{pmatrix} w_{11} & 0 & \ldots & 0 \\ 0 & w_{22} & \ldots & 0 \\ \vdots & \vdots & \ddots & \vdots \\ 0 & 0 & \ldots & w_{kk} \end{pmatrix} \quad (4)$$

where the top-left submatrix $\Lambda_{11} \in \mathbb{R}^{k \times k}$ is a learnable, strictly diagonal matrix.

This diagonal structure enforces a direct, one-to-one mapping, ensuring each of the $k$ perturbed genes directly influences only one of the first $k$ learned meta-pathways ( Fig. 2-**A**). The bottom-left zero block, $\mathbf{0}_{(d-k) \times k}$, is critical for preventing intervention spillover by explicitly blocking the influence of perturbed genes on the remaining learned meta-pathways. The other submatrices, $\boldsymbol{W}_{12}$ and $\boldsymbol{W}_{22}$, are learnable and dense, allowing the model flexibility to discover complex relationships between unperturbed genes and all learned meta-pathways. Crucially, this preconditioned structure allows the intervention mask $\boldsymbol{m}$ to be predefined and fixed rather than learned, directly satisfying the theoretical requirements for identifiability.

**Latent grouping via an interaction block (Fig. 2-B).** To model the complex non-linear dynamics of biological systems, our architecture extends this preconditioning approach by learning multiple distinct subgraph representations for each (exogenous) latent factor. An intermediate *Interaction Block* then processes these subgraph activations, allowing the model to capture higher-order dependencies by learning unique interaction patterns among the subgraphs within each learned latent factor (Fig. 2-**B**).

**Exogeneous latent factor inference via VAE (Fig. 2-C).** Next, an aggregation layer combines these processed features to compute the parameters ($\mu_{\mathbf{z}}$ and $\sigma_{\mathbf{z}}$) for the (exogenous) latent distributions (Fig. 2-**C**). Thus, the final stochastic representations of the latent factors are then obtained using the

reparameterization trick $\mathbf{z} = \mu_{\mathbf{z}} + \sigma_{\mathbf{z}} \odot \epsilon; \quad \epsilon \sim \mathcal{N}(\mathbf{0}, \mathbf{I})$. Subsequently, a latent modulator predicts a conditional shift for the targeted latent variables, strictly modifying only the specific latent variable associated with the perturbation while preserving the independence of the others.

**Causal Latent factor (causal meta-pathway) inference via DAGMA (Fig. 2-D, Sec. 3.2)**. To capture the downstream biological consequences, a DAGMA layer propagates these atomic interventions through a learned directed acyclic graph, transforming the independent noise terms into causally dependent pathway activities. Finally, a non-linear dense decoder maps these propagated causal states back to the high-dimensional gene expression space to predict the final post-perturbation phenotype.

For a comprehensive architectural description, we refer the reader to Sec. E.2.

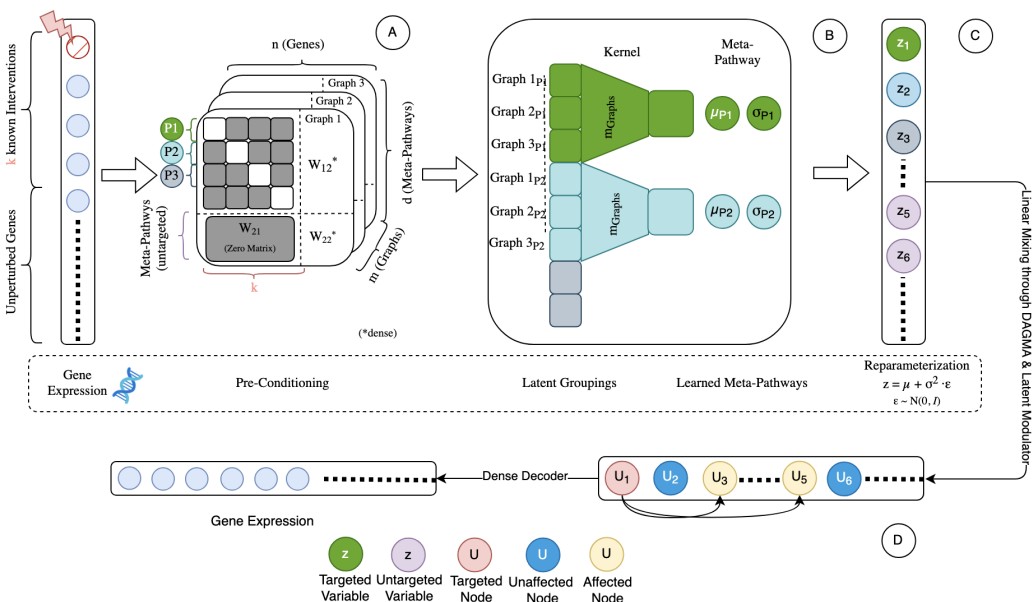

Figure 2: **The GraphPathway Encoder architecture of RAPTORGraph. A**: Gene expression of control cells ($\mathbf{z}_{\text{obs}}$) is fed as input to the preconditioned linear layer, which enforces a sparse, one-to-one mapping for known perturbed genes to explicitly prevent intervention spillover. **B**: Complex dependencies are captured by learning multiple parallel graph representations, which are then aggregated to model non-linear interactions. **C**: A standard VAE reparameterization head generates the stochastic latent variables ($\mathbf{z}$). **D**: A Latent Modulator takes the perturbation label as input and applies atomic perturbations ($\Delta$) to the independent latents ($\mathbf{z}$), shifting the distribution from the observed latent state ($\mathbf{z}_{\text{obs}}$) to the interventional latent state ($\mathbf{z}_{\text{int}}$). Both $\mathbf{z}_{\text{obs}}$ and $\mathbf{z}_{\text{int}}$ are then transformed by the DAGMA layer through the learned DAG via linear mixing to yield the final causal states ($\mathbf{u}$), which are reconstructed into the predicted control ($\hat{\mathbf{x}}_{\text{obs}}$) and perturbed ($\hat{\mathbf{x}}_{\text{int}}$) gene expression profiles by a dense decoder.

## 3.2 Deep Causal Graph Learning with DAGMA

The GraphPathway Encoder with preconditioning fundamentally alters the causal discovery task. By fixing the mapping from known interventions to specific latent variables, it eliminates the need for permutation-based discovery methods that assume a dense, entangled encoder. However, this fixed ordering makes simple acyclicity constraints, such as enforcing an upper-triangular adjacency matrix, unsuitable. These methods impose a strict and arbitrary causal hierarchy, as the influence of one latent variable on another is entirely restricted by their ordering (e.g., latent $i$ can only influence latent $j$ if $i < j$), which would conflict with the true biological relationships we aim to discover. For this, we adapt the DAGMA framework (Bello et al., 2022) for use in a deep learning context.

**DAGMA** reformulates the combinatorial search for a DAG into a continuous optimization problem. This is achieved by minimizing a score function that measures data fit, subject to a differentiable constraint that is satisfied only if the graph is acyclic. The DAGMA acyclicity constraint, $h_s(\boldsymbol{A})$, is

based on a log-determinant function:

$$h_s(\boldsymbol{A}) = -\log \det(s\boldsymbol{I} - \boldsymbol{A} \circ \boldsymbol{A}) + d \log s = 0 \tag{5}$$

where $\boldsymbol{A} \in \mathbb{R}^{d \times d}$ is the weighted adjacency matrix of the latent graph, $s > 0$ is a scaling scalar, and the condition holds if and only if $\boldsymbol{A}$ represents a DAG. The optimization then solves a sequence of unconstrained problems that jointly minimize a data-fit score and the acyclicity penalty.

**DAGMA implementation.** We implement a modified DAGMA approach designed to learn the invariant, baseline causal structure of the biological system. To ensure stability and manage the trade-off between reconstruction fidelity and acyclicity, we introduce two critical regularization. First, we employ a path-following schedule where the score weight $\mu$ is gradually decreased (see Eq. (7)), allowing the model to prioritize feature learning early in training before strictly enforcing the graph constraint. Second, we isolate gradient flow by computing $\mathcal{L}_{\text{DAGMA}}$ on detached latent representations. This ensures the acyclicity constraint optimizes the graph structure without distorting the encoder's feature extraction. A detailed description of the DAGMA layer, its objective, theoretical background, and implementation can be found in Sec. E.3.

### 3.3 TRAINING OBJECTIVE

The model is trained end-to-end by minimizing a composite objective function, $\mathcal{L}_{\text{total}}$, that integrates four distinct components, each tailored to a specific aspect of the learning problem. The overall objective is a weighted sum of these individual losses:

$$\mathcal{L}_{\text{total}} = \alpha_{\text{rec}} \mathcal{L}_{\text{MSE}} + \beta_{\text{KL}} \mathcal{L}_{\text{KL}} + \gamma_{\text{MMD}} \mathcal{L}_{\text{MMD}} + \delta_{\text{DAGMA}} \mathcal{L}_{\text{DAGMA}} \tag{6}$$

where $\alpha_{\text{rec}}, \beta_{\text{KL}}, \gamma_{\text{MMD}}$, and $\delta_{\text{DAGMA}}$ are scalar hyperparameters that balance the contribution of each term.

**Reconstruction Loss ($\mathcal{L}_{\text{MSE}}$).** For observational data, the model is trained to faithfully reconstruct the input gene expression profiles. This is measured by the Mean Squared Error (MSE) between the decoder's output and the original unperturbed input, ensuring the model learns a high-fidelity representation of the basal cellular state.

**KL Divergence ($\mathcal{L}_{\text{KL}}$).** As a standard component of a VAE, a Kullback-Leibler divergence term is included to regularize the latent space. It encourages the learned posterior distribution over the latent variables, $q(\mathbf{z}|\mathbf{x})$, to be close to a prior distribution, $p(\mathbf{z})$, which is typically a standard normal distribution. This promotes a smooth and well-structured latent space.

**Interventional Prediction Loss ($\mathcal{L}_{\text{MMD}}$).** For interventional data, the model's ability to predict out-of-distribution outcomes is evaluated using the Maximum Mean Discrepancy (MMD). This loss measures the distributional distance between the model-predicted cell states following a perturbation and the ground-truth distribution of experimentally observed cells for that same perturbation.

**Causal Graph Loss ($\mathcal{L}_{\text{DAGMA}}$).** The DAGMA loss is computed on the latent representations of observational data to learn the invariant causal graph structure between the learned meta-pathways. This objective integrates three key components: a data-fit score, an L1 penalty to encourage sparsity, and the differentiable acyclicity constraint. The loss is defined as:

$$\mathcal{L}_{\text{DAGMA}} = \mu(S(\boldsymbol{A}; \boldsymbol{X}) + \lambda_1 \|\boldsymbol{A}\|_1) + h(\boldsymbol{A}) \tag{7}$$

where $S(\boldsymbol{A}; \boldsymbol{X})$ is a score (see Sec. E.3.2), $\|\boldsymbol{A}\|_1$ is the L1 norm of the adjacency matrix, and $h(\boldsymbol{A})$ is the acyclicity penalty. The hyperparameters $\mu$ and $\lambda_1$ control the weighting of the score and sparsity terms, respectively.

## 4 EXPERIMENTAL SETTINGS

### 4.1 OPTIMAL TRANSPORT FOR DETERMINISTIC INTERVENTIONAL LEARNING

A fundamental challenge in learning from single-cell perturbation data is the absence of a direct one-to-one mapping between control and perturbed cell populations, as it is impossible to observe the same cell before and after the intervention. This makes learning the treatment effect

a fundamental counterfactual problem, where the central challenge is to identify a general transformation that maps the control distribution to the perturbed distribution in the absence of paired examples. To address this counterfactual gap, many models create training pairs by randomly matching control and interventional samples. While computationally efficient, this random pairing is a primary cause of what we term *mean collapse*: a process where the model learns the average response, collapsing the distribution towards its mean and eroding cellular diversity (Fig. 3).

To address these issues, we employ a Wasserstein-based OT (Cuturi, 2013; Flamary et al., 2021) framework to globally pair control cells with perturbed cells under unique conditions (Fig. 3b). Although several recent methods address unpaired single-cell perturbation data (Bunne et al., 2023; Dong et al., 2023), we adopt OT as an offline pairing step for simplicity while avoiding mean collapse. See Sec. E.4 for more details regarding OT. Extended results, including empirical evidence supporting the necessity of OT and a comprehensive analysis of OT complexity, are detailed in Sec. F.4 and Sec. F.3, respectively.

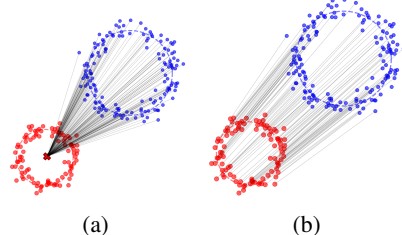

(a)          (b)

Figure 3: Illustrative example of mean collapse problem. (a) Random pairing collapses the source distribution towards the target mean. (b) The OT preserves the underlying population structure.

### 4.2 EVALUATION SETUP AND METRICS

To ensure a consistent and fair benchmark across all models taken into account, we use Multi-Kernel Maximum Mean Discrepancy (MK-MMD) to assess the similarity between predicted and true gene expression distributions. The MK-MMD is calculated per-condition and then averaged to provide an overall measure of distribution similarity. Alongside objective metric comparison, we use Precision@10 to evaluate non-additive interactions of combinatorial perturbations (see Sec. D).

We used the Perturb-seq single-cell RNA-seq dataset curated by Norman et al. (2019) as processed in the CPA study (Lotfollahi et al., 2023a) and refer to it as the Norman-CPA dataset. It profiles 105 gene perturbations in K562 cells, including both single and double perturbations. In total, 284 conditions across ∼108,000 cells were measured, of which 131 represent unique gene–gene combinations and the remainder single perturbations or controls, respectively. We utilize the entire set of single-perturbations during training and evaluate the subsequent methods on the double-perturbations.

Additionally, we used the CRISPRi single-cell RNA-seq dataset (exp6) curated by Replogle et al. (2020) as processed in the PerturBase study (Wei et al., 2024) and refer to it as the Replogle2020 dataset. This dataset profiles 82 gene perturbations in K562 cells, including both single (44), double perturbations (37) and control. The 82 conditions were measured across ∼22,740 cells.

## 5 RESULTS

To first establish the foundational accuracy of RAPTORGraph, we benchmarked its generative performance against four state-of-the-art models: SENA, scGPT, dVAE, and GEARS. We assessed two key aspects of prediction fidelity primarily using the single-cell perturbation dataset from Norman et al. (2019). We prioritized this dataset for the comprehensive comparison because the implementations of several baseline models have auxiliary files and preprocessing steps that are tightly coupled to it, making adaptation to other datasets non-trivial. However, to further demonstrate generalizability, we provide an additional evaluation on the Replogle2020 dataset in Sec. D.3, restricted to a comparison between RAPTORGraph and scGPT.

To validate our hyperparameter selection, we conducted a comprehensive ablation study on the $\beta$-VAE regularization parameter ($\beta$) and the DAGMA loss weights ($\mu, \lambda_1$). Our analysis reveals a critical trade-off between distribution matching and the preservation of biological heterogeneity. While higher regularization ($\beta \geq 0.8$) achieves competitive MK-MMD scores, it induces severe posterior collapse, with the variance of predicted control cell states dropping by an order of magnitude compared to the optimal setting ($10^{-4}$ vs $10^{-3}$). Conversely, lower regularization ($\beta = 0.1$) preserves higher variance but results in unstable training and significantly higher MK-MMD. We identified $\beta \approx 0.4$ as the optimal operating point, minimizing MK-MMD while maintaining a latent KL divergence ($\approx 10^{-3}$) consistent with established baselines (dVAE and SENA), thereby ensuring the model

captures meaningful cellular diversity rather than regressing to the population mean. Detailed results of the full hyperparameter sweep are provided in Sec. F.6.

**Reconstruction capabilities.** First, we measured the MSE between the ground truth and predicted profiles of control cells to evaluate how well each model reconstructs the basal, unperturbed cellular state. Second, we calculated the MK-MMD between the distributions of ground truth and predicted profiles for double-gene perturbations to assess the model's ability to capture the global transcriptional shift following a complex intervention. A detailed description of these metrics is provided in Sec. D.

The results in Table 1 highlight a crucial trade-off between reconstruction fidelity (MSE) and distributional accuracy (MK-MMD). RAPTORGraph achieves a strong overall balance, demonstrating highly competitive performance across both reconstruction and generative capabilities.

| Metric | SENA | scGPT | dVAE | GEARS | RAPTORGraph (ours) |
|---|---|---|---|---|---|
| MK-MMD | $0.522 \pm 0.003$ | $0.176 \pm 0.001$ | $\mathbf{0.076 \pm 0.000}$ | –* | *0.112 ± 0.001* |
| MSE | $\mathbf{0.043 \pm 0.000}$ | $0.066 \pm 0.000$ | $0.062 \pm 0.000$ | –* | *0.045 ± 0.000* |

Table 1: Results using the Norman-CPA dataset are averaged over 3 runs. Bold: best results; Italics: second best. Values are mean $\pm$ variance. *We excluded the GEARS model because it behaves in a fundamentally different way that is incompatible with our evaluation framework, making a fair and direct comparison impossible (see Sec. D.7).

**Prediction of non-additive genetic perturbations.** This complex and biologically significant benchmark assesses a model's ability to predict emergent effects that are not simple linear combinations of single-gene perturbations (Table 2). Specifically, we compared the models based on their ability to predict five distinct interaction types, using Precision@10 to quantify how effectively each model could identify true interactions within its top 10 predictions, a direct measure of its utility for experimental discovery (see Sec. D.4).

| Metric | dVAE | SENA | GEARS | scGPT | RAPTORGraph (ours) |
|---|---|---|---|---|---|
| Synergy | $0.000 \pm 0.000$ | $0.233 \pm 0.153$ | $0.333 \pm 0.115$ | $0.100 \pm 0.000$ | $\mathbf{0.367 \pm 0.058}$ |
| Suppression | $0.000 \pm 0.000$ | $0.167 \pm 0.115$ | $0.333 \pm 0.115$ | $\mathbf{0.400 \pm 0.000}$ | $0.267 \pm 0.153$ |
| Neomorphism | $0.333 \pm 0.153$ | $0.333 \pm 0.115$ | $0.133 \pm 0.058$ | $0.100 \pm 0.000$ | $\mathbf{0.433 \pm 0.058}$ |
| Redundancy | $\mathbf{0.933 \pm 0.115}$ | $0.467 \pm 0.058$ | $0.567 \pm 0.115$ | $0.100 \pm 0.000$ | $0.800 \pm 0.100$ |
| Epistasis | $0.567 \pm 0.153$ | $\mathbf{0.800 \pm 0.100}$ | $0.533 \pm 0.115$ | $0.400 \pm 0.000$ | $0.433 \pm 0.115$ |

Table 2: Comparison of non-additive genetic interaction prediction. Precision@10 scores for five baseline models across distinct genetic interaction subtypes on the Norman et al. dataset. Higher is better. Values are mean $\pm$ variance over 3 runs.

We further investigated the contribution of OT to this performance. Our ablation studies reveal that OT preprocessing is a key driver of the model's ability to predict complex non-additive interactions, particularly redundancy and epistasis, significantly outperforming a random pairing baseline. Importantly, this benefit comes with negligible computational overhead ($\approx 2.2\%$ of total training time). Detailed analyses of these empirical benefits and the computational cost are provided in Sec. F.5 and Sec. F.3, respectively.

**Reverse perturbation analysis.** While predicting genetic interactions demonstrates a model's forward predictive power, a more profound test is its ability to reverse-engineer the genetic cause of an observed phenotype. To this end, we challenged a model to identify the specific genetic perturbation responsible for a given cellular gene expression profile. The analysis was performed on a controlled combinatorial space of 20 transcription factors from the Norman et al. (2019) dataset. For each model, we generated a database of predicted expression profiles for all possible double-gene perturbations within this subset. We then used the ground truth mean expression profiles from experimentally observed double perturbations as queries, ranking the database entries by similarity to identify the most likely causal perturbation.

We first evaluated the models on the highly stringent task of identifying the exact causal gene combination, quantified by the Correct (Exact Match) Hit Rate @ K. As shown in Fig. S5, all models achieve modest hit rates, underscoring the profound difficulty of this reverse-engineering challenge. However, a more practical measure of a model's utility is its ability to identify biologically influential genes. We assessed this using the Relevant (Gene Overlap) Hit Rate @ K, which credits a model for retrieving perturbations containing at least one of the correct genes (see Fig. S7 and Table 3). RAPTORGraph demonstrates exceptional performance, consistently ranking among the top models. This highlights that while pinpointing exact higher-order interactions is an ongoing challenge, RAPTORGraph is highly effective at identifying the key genetic players driving a cellular response, making it a valuable tool for hypothesis generation and experimental design.

| $k$ | dVAE | SENA | GEARS | scGPT | RAPTORGraph (ours) |
|---|---|---|---|---|---|
| 1 | $0.333 \pm 0.115$ | $0.244 \pm 0.019$ | $\mathbf{0.344 \pm 0.051}$ | $0.233 \pm 0.000$ | $0.294 \pm 0.059$ |
| 3 | $0.556 \pm 0.051$ | $0.444 \pm 0.069$ | $0.578 \pm 0.051$ | $0.533 \pm 0.000$ | $\mathbf{0.608 \pm 0.122}$ |
| 5 | $0.733 \pm 0.088$ | $0.600 \pm 0.058$ | $0.722 \pm 0.077$ | $0.700 \pm 0.000$ | $\mathbf{0.784 \pm 0.090}$ |

Table 3: Hit Rate @ K (relevant) for the reverse perturbation task. The metric measures the fraction of queries where at least one correct gene was identified in the top K predictions. Higher values are better. Values are mean $\pm$ variance over 3 runs.

**RAPTORGraph learns biologically meaningful causal latent factors.** To confirm that RAPTOR-Graph learns biologically meaningful structures rather than arbitrary representations, we examined the functional identity of the latent factors using Gene Set Enrichment Analysis (GSEA). We utilized an in-silico perturbation approach to generate transcriptional signatures for each latent factor and tested them against the MSigDB Hallmark collection (Liberzon et al., 2015). We observed a robust alignment between the known biological roles of perturbed genes and the pathway enrichments of their corresponding latent factors. For instance, the latent factor constrained to the stress-response transcription factor *JUN* displayed strong negative enrichment for cell cycle signatures ("E2F Targets," FDR $< 0.001$), consistent with its role in stress-induced growth arrest. Similarly, the factor associated with the tumor suppressor *TP73* showed significant negative enrichment for "G2-M Checkpoint" targets (FDR $< 0.03$). Furthermore, the factor linked to the differentiation regulator *EGR1* exhibited massive downregulation of E2F targets (FDR $\approx 0.0$), consistent with the cell cycle exit necessary for differentiation. These results demonstrate that the model successfully disentangles complex transcriptomic profiles into interpretable, functionally coherent modules. A detailed description of the methodology is provided in Sec. D.6 and a comprehensive heatmap and analysis of pathway enrichments can be found in Sec. F.2

## 6 CONCLUSION

Developing interpretable *in silico* models is crucial for therapeutic discovery, yet current approaches face two key challenges: intervention spillover, where dense encoders create entangled latent signals that confound causal discovery, and mean collapse, where the absence of one-to-one cell pairings erodes cellular diversity. We introduce RAPTORGraph, a framework that directly resolves these issues. To counteract intervention spillover, it combines a preconditioned GraphPathway layer, enforcing the sparse mappings required for clean single-node interventions, with a DAGMA layer for robust DAG discovery. This architectural design eliminates the artifactual entanglement of input genes with latent pathways, while the subsequent DAG module explicitly learns the legitimate biological causal relationships and correlations between the resulting sparse pathway activities. To prevent mean collapse, RAPTORGraph globally pairs control and perturbed populations through OT. RAPTORGraph not only achieves a superior balance between reconstruction fidelity and distributional accuracy compared to state of the arts, but also demonstrates a superior capability to predict complex, non-additive genetic interactions, and can reverse-engineer genetic drivers to generate testable hypotheses. RAPTORGraph bridges predictive power and mechanistic insight, enabling not only prediction a perturbation's effect but also understand why. Future work will focus on deploying RAPTORGraph for the hypothesis-driven discovery of novel drug targets and exploring its utility in uncovering cell-type specific pathway mechanisms.

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

## A  Nomenclature and Mathematical Notation

In this section, we provide a comprehensive list of the mathematical notations used throughout our work, organized by category. This serves as a centralized reference to ensure clarity and consistency.

### A.1  General Variables and Data Structures

We begin by defining the fundamental variables and data structures used to represent observed and latent spaces.

| Symbol | Description |
| --- | --- |
| $\mathbb{R}$ | The set of real numbers. |
| $\mathbf{x}$ | A vector of observed variables (e.g., gene expression). |
| $\mathbf{z}$ | A vector of general latent variables in the learned representation space. |
| $\mathbf{e}$ | A vector of general, unspecified noise terms. |
| $\boldsymbol{X}$ | A matrix representing a set of observed data samples. |
| $\boldsymbol{I}$ | The identity matrix. |

### A.2  Causal Models and Graphs

Next, we define the components of the Structural Causal Model (SCM) that describes the underlying data-generating process.

| Symbol | Description |
| --- | --- |
| $\mathcal{G}$ | A DAG representing the causal structure. |
| $\boldsymbol{A}$ | The weighted adjacency matrix of the causal graph $\mathcal{G}$. |
| $\boldsymbol{B}$ | A general adjacency matrix. |
| $\boldsymbol{W}$ | A general weight matrix. |
| $\mathrm{Pa}_{\mathcal{G}}(\cdot)$ | A function that returns the parent nodes of a given node in $\mathcal{G}$. |
| $\bar{\mathcal{G}}$ | The transitive closure of the graph $\mathcal{G}$. |
| $\mathbf{u}$ | The true, unobserved latent causal variables in the SCM. |
| u | A single latent causal variable. |
| $\mathbf{z}$ | A vector of exogenous noise terms for the SCM. |
| $f_i$ | The structural causal function for a single variable $u_i$. |
| $\mathrm{Tr}(\cdot)$ | The trace function for a matrix. |

### A.3  Model Functions and Representations

We denote the core functions of our model, which learn to map between the observed and latent spaces, as follows.

| Symbol | Description |
| --- | --- |
| $g^*$ | The true, unknown data-generating (mixing) function. |
| $g$ | The learned decoder network. |
| $f^*$ | The ideal, oracle encoder function. |
| $f_\theta$ | The practical, parameterized encoder network. |
| $\hat{\mathbf{z}}$ | The learned latent representation, output of the encoder ($f_\theta(\mathbf{x})$). |
| $\hat{\mathbf{x}}$ | A single data sample generated by the model's decoder. |
| $\hat{\boldsymbol{X}}$ | A matrix representing a batch of data samples generated by the model's decoder. |
| $a$ | The composed function $f_\theta \circ g^*$. |
| $\tilde{\mathbf{z}}$ | The true latent representation produced by the $f^*$. |
| $\Delta\tilde{\mathbf{z}}$ | The ideal, single-node change in the latent space. |
| $\Delta\mathbf{z}$ | The approximated (dense) change in the latent space. |
| $\mathcal{N}$ | The normal (Gaussian) distribution. |

### A.4 INTERVENTION-SPECIFIC VARIABLES

Here, we list the variables specifically related to the modeling of interventions.

| Symbol | Description |
|--------|-------------|
| $q_{\text{int}}$ | The intervention encoder network. |
| $\boldsymbol{i}$ | The one-hot vector specifying the target of an intervention. |
| $\boldsymbol{c}$ | A general condition vector input to an intervention network. |
| $\boldsymbol{m}$ | The mask vector specifying an intervention's target(s). |
| $c$ | The scalar strength of a latent intervention. |
| $\mathbf{z}_{\text{obs}}$ | Latents from an observational (control) sample. |
| $\mathbf{z}_{\text{int}}$ | Latents from an interventional sample. |
| $P_Z^{(i)}$ | The interventional distribution over latent variables. |
| $I_k$ | The set of intervened variables in environment $k$. |
| $\boldsymbol{t}_i$ | The targets of an intervention for a variable $i$. |
| $\boldsymbol{u}$ | An auxiliary variable providing conditioning information (e.g., environment index). |

### A.5 IDENTIFIABILITY AND TRANSFORMATIONS

We define symbols used in the context of identifiability proofs and the resulting transformation classes.

| Symbol | Description |
|--------|-------------|
| $\sim_T$ | Equivalence class for a transformation T. |
| $\sim_C$ | Equivalence class for a transformation C. |
| $\mathbf{A}$ | The matrix component of an affine transformation. |
| $\mathbf{c}$ | The vector component of an affine transformation. |
| $e$ | A scalar factor in a scaling transformation. |
| $b$ | An offset in a shift transformation. |

### A.6 PROBABILITY, LOSSES, AND FUNCTIONS

We define the probability functions, loss functions, and mathematical operators used to train our models and enforce constraints.

| Symbol | Description |
|--------|-------------|
| $\mathrm{P}(\cdot)$ | PMF (for discrete variables). |
| $\mathrm{p}(\cdot)$ | PDF (for continuous variables). |
| $\mathrm{F}(\cdot)$ | CDF. |
| $p_{\text{data}}$ | The true, underlying data-generating distribution (PDF). |
| $p_{\text{model}}$ | The learned distribution of the model (PDF). |
| $\mathbb{E}[\cdot]$ | The expectation function. |
| $\mathcal{L}_{\text{DAGMA}}$ | Loss function related to the causal graph structure. |
| $\mathcal{L}_{\text{MMD}}$ | Loss function based on the MMD. |
| $h$ | A function used to enforce graph acyclicity. |
| $S$ | A score function for graph discovery. |
| $\mathrm{MMD}(\cdot, \cdot)$ | The MMD function, which computes the distance between two distributions. |
| $\mathrm{MK\text{-}MMD}(\cdot, \cdot)$ | The MK-MMD function, a variant of MMD using multiple kernels. |
| $k(\cdot, \cdot)$ | A kernel function used in MMD computation. |

### A.7 HYPERPARAMETERS

We list the key hyperparameters that control our model's training and behavior.

| Symbol | Description |
|---|---|
| $\lambda_{\text{score}}$ | Regularization hyperparameter for a graph score. |
| $\sigma_{\text{MMD}}$ | The bandwidth hyperparameter for the Gaussian RBF kernel. |
| $\beta$ | The weights for the convex combination in Multi-Kernel MMD. |
| $L$ | The number of kernels used in the MK-MMD calculation. |
| $m$ | The number of graphs. |
| $i$ | The number of samples in a batch of observed data. |
| $j$ | The number of samples in a batch of predicted data. |

## A.8 DOWNSTREAM ANALYSIS METRICS

We define variables used in the downstream evaluation tasks, such as the genetic interaction analysis.

| Symbol | Description |
|---|---|
| $\Delta$ | The perturbation effect vector (mean of perturbed minus mean of control). |
| $\Delta(A + B)_{\text{naive}}$ | The naive additive effect vector, defined as $\Delta A + \Delta B$. |
| $\|\cdot\|_2$ | The L2 norm of a vector. |
| $\cos(\cdot, \cdot)$ | The cosine similarity between two vectors. |
| $\bar{\mathbf{x}}$ | The mean expression profile of a cell population. |
| $N$ | The number of cells for a given perturbation condition. |
| $f_M$ | The prediction function for a given model $M$. |
| Synergy Ratio | The score used to measure synergy and suppression. |
| Neomorphism Score | The score used to measure neomorphism. |
| Redundancy Score | The score used to measure redundancy. |
| Epistasis Score | The score used to measure epistasis. |

## A.9 DIMENSIONALITY AND SPACES

Finally, we list symbols for dimensionality and the properties of the data spaces.

| Symbol | Description |
|---|---|
| $n$ | The dimensionality of the observed data space. |
| $d$ | The dimensionality of the latent space. |
| $d$ | The number of learned pathways, equivalent to $d$. |
| $p$ | The degree of a polynomial mixing function. |
| $\mathcal{Z}$ | The support of the latent distribution. |
| $\mathcal{X}$ | The support of the observed distribution. |
| $n$ | The number of genes in the expression profile, equivalent to $n$. |

# B MATHEMATICAL PROOFS AND THEORETICAL BACKGROUND

This appendix provides a detailed theoretical examination of the core challenges in causal representation learning that motivate our proposed architecture. We begin by formally defining and proving the incompatibility of dense encoders via the Intervention Spillover Problem, and then discuss the role of prior knowledge in differentiating causal modeling strategies.

## B.1 THE INTERVENTION SPILLOVER PROBLEM

This section provides the formal theoretical background for the *Intervention Spillover Problem*. We formally define this problem, arguing that it is a direct consequence of the inherent properties of dense encoder architectures commonly used in causal representation learning (CRL). We then prove the fundamental incompatibility between these architectures and the requirements for causal identifiability.

**Definition B.1** (The Intervention Spillover Problem). The *Intervention Spillover Problem* is the phenomenon where a targeted single-variable intervention in a high-dimensional observation space is transformed into a dense, multi-node signal in the lower-dimensional latent space. This issue arises as a direct consequence of using a practical, dense encoder architecture, which inevitably corrupts the intervention's sparsity and creates a fundamental incompatibility with CRL methods that require sparse, single-node latent interventions for identifiability.

### B.1.1 MATHEMATICAL PRELIMINARIES AND SETUP

While the problem described is general, this analysis formalizes it within the context of a *linear SCM* for clarity. Let the gene expression space be $\mathbb{R}^n$ and the latent space be $\mathbb{R}^d$, with $d \ll n$. Let $f_\theta : \mathbb{R}^n \to \mathbb{R}^d$ be the practical, differentiable encoder network we train. It maps an observation vector $\mathbf{x}$ to an *approximated* latent vector $\mathbf{z} = f_\theta(\mathbf{x})$. The local behavior of this encoder is described by its Jacobian matrix $\boldsymbol{J}_\theta(\mathbf{x}) \in \mathbb{R}^{d \times n}$.

We model the latent space with a linear SCM, where causal relationships are defined by an adjacency matrix $\boldsymbol{A}$. An intervention on a single gene creates a post-intervention vector $\mathbf{x}_{\text{int}}$ from a corresponding control vector $\mathbf{x}_{\text{obs}}$. This experimental paradigm is characteristic of modern pooled CRISPR screens (e.g., Perturb-seq), where the genetic target of each intervention is known by design.

### B.1.2 FORMAL ASSUMPTIONS

**Assumption B.2** (Ideal Causal Encoder and Single-Node Intervention Target). An *Ideal Causal Encoder*, $f^*$, is an unknown oracle function that perfectly disentangles the effect of any intervention. The key property is that the *true* latent change, $\Delta \tilde{\mathbf{z}} = f^*(\mathbf{x}_{\text{int}}) - f^*(\mathbf{x}_{\text{obs}})$, corresponds to a single-node perturbation. This ideal latent change is defined as $\Delta \tilde{\mathbf{z}} = \boldsymbol{m} \cdot c$, where $c$ is the latent intervention strength and the target mask $\boldsymbol{m} \in \{0, 1\}^d$ is *one-hot*:

$$\sum_{i=1}^{d} m_i = 1 \tag{S1}$$

In a *closed-world experimental setting*, where the targeted gene is known *a priori*, the behavior of the $f^*$ must fulfill the single-node intervention requirement. The following lemma formalizes this ideal mapping.

**Lemma B.3** (Equivalence in Ideal Latent Space). *For an Ideal Causal Encoder $f^*$ operating in a closed-world setting, the latent representation of the post-intervention sample is equivalent to the latent representation of the control sample plus the ideal single-node latent perturbation. Formally:*

$$f^*(\mathbf{x}_{int}) = f^*(\mathbf{x}_{obs}) + \boldsymbol{m} \cdot c \tag{S2}$$

*Proof.* The proof follows directly from the definition of the ideal latent change in theorem B.2. We start with the identity $\Delta \tilde{\mathbf{z}} = f^*(\mathbf{x}_{\text{int}}) - f^*(\mathbf{x}_{\text{obs}})$. Substituting the single-node perturbation gives $f^*(\mathbf{x}_{\text{int}}) - f^*(\mathbf{x}_{\text{obs}}) = \boldsymbol{m} \cdot c$. Rearranging the terms yields the desired result. □

**Assumption B.4** (Practical Model Approximation and Dense Jacobian). Our *Practical Model* uses an encoder $f_\theta(\mathbf{x})$ and a perturbation inference network $q_{\text{int}}$. The underlying encoder network has a *dense Jacobian matrix* $\boldsymbol{J}_\theta(\mathbf{x})$.

*Remark* B.5 (Architectural Bias vs. Learned Function). It is important to distinguish between the architecture of the encoder and the function it learns. While it is theoretically possible for a densely-connected network to learn an effectively sparse Jacobian, the architecture itself possesses a strong inductive bias towards dense mappings. Standard training objectives provide no direct incentive for learning a sparse function.

### B.1.3 MAIN RESULT: THE INCOMPATIBILITY

**Lemma B.6** (Incompatibility of the Practical Model). *Given the practical model's reliance on a dense encoder network (theorem B.4), there is no guarantee that an intervention on a single gene will produce a single-node latent intervention target vector $\boldsymbol{m}$ as required by the ideal model in theorem B.2.*

*Proof.* The proof rests on the direct conflict between the output of the practical encoder and the requirements of the ideal model in a closed-world setting. In this setting, we have access to both the control sample $\mathbf{x}_{\text{obs}}$ and the post-intervention sample $\mathbf{x}_{\text{int}}$. Our trained practical encoder, $f_\theta$, is a known deterministic function (e.g., a trained MLP).

When we process these known samples, the encoder produces an approximated latent change:

$$\Delta \mathbf{z} = f_\theta(\mathbf{x}_{\text{int}}) - f_\theta(\mathbf{x}_{\text{obs}}) \tag{S3}$$

Due to the dense Jacobian of $f_\theta$ (theorem B.4), each component of the latent vector $\mathbf{z}$ is a function of all input components in $\mathbf{x}$. Consequently, a change in a single gene will result in a change across nearly all latent dimensions. The resulting vector $\Delta \mathbf{z}$ is therefore dense.

The objective for a conditional network $q_{\text{int}}$ is to learn to predict this latent change. For the model to perfectly capture the intervention effect, the learned perturbation $(\boldsymbol{m} \cdot c)$ must match the observed latent change. This implies minimizing the loss:

$$\mathcal{L} = ||\Delta \mathbf{z} - (\boldsymbol{m} \cdot c)||^2 \tag{S4}$$

To achieve a loss of zero, the learned perturbation $(\boldsymbol{m} \cdot c)$ must be equal to the dense vector $\Delta \mathbf{z}$. This forces the learned target mask $\boldsymbol{m}$ to be dense, which directly contradicts the one-hot requirement for a single-node intervention as defined in Eq. (S1). Therefore, the practical model's architecture is fundamentally misaligned with the goal of identifying single-node latent interventions. $\square$

### B.1.4 IMPLICATIONS FOR CAUSAL DISCOVERY IN DENSE ENCODERS

This mathematical reality creates a cascade of problems for causal discovery models that rely on a learned intervention network to identify the latent target of a perturbation. The core of the issue is the need to make a discrete, single-target selection. This is typically addressed using a temperature-scaled softmax function, a mechanism that mimics the Gumbel-Softmax trick for differentiable categorical sampling (Jang et al., 2016). Given the output logits, $\boldsymbol{o}$, from an intervention network, the probability for each target is calculated as:

$$\pi_i = \frac{\exp(\boldsymbol{o}_i/\tau)}{\sum_{k=1}^{d} \exp(\boldsymbol{o}_k/\tau)} \tag{S5}$$

where $\tau > 0$ is a temperature hyperparameter that controls the sharpness of the output distribution.

As the temperature $\tau \to 0$, the softmax output approaches a discrete, one-hot vector (a proxy for a Dirac delta function), representing a single, "hard" choice. However, this creates a fundamental contradiction when applied to the output of a dense encoder. The encoder produces a dense latent change vector, $\Delta \mathbf{z}$, where the evidence suggests a multi-node perturbation. To satisfy the model's objective of selecting a single target, a low temperature $\tau$ must be used to force the softmax into a "winner-take-all" state. This forces the model to ignore the dense evidence and declare a single winner, which can lead to it learning spurious, causally unfaithful mappings.

The temperature-scaled softmax is therefore caught in an architectural dissonance: it is a mechanism forced to make a discrete choice that fundamentally misrepresents the dense, continuous signal it receives from the encoder.

Furthermore, in the closed-world setting of a Perturb-seq experiment, the mapping from a perturbed gene to its corresponding latent variable is known beforehand. Therefore, the task of "exploratory perturbation" (i.e., discovering which latent variable was the target) is not actually necessary. The true challenge is to learn the *scale* or *power* of the intervention's effect on the causal representation, not its target, a task for which these architectures are ill-suited.

### B.2 THE ROLE OF PRIOR KNOWLEDGE IN CAUSAL MODELING

The theoretical gaps in existing models can be understood by positioning them on a spectrum between two complementary modes of scientific inquiry: exploratory and confirmatory research (Herrmann et al., 2024). Exploratory research is concerned with hypothesis generation; it involves analyzing data to uncover novel patterns, structures, and relationships without a strong pre-existing theory. In contrast, confirmatory research is concerned with hypothesis testing; it begins with a specific, pre-formulated hypothesis and uses data to determine if that hypothesis is supported or refuted. This distinction helps clarify the intended purpose and inherent trade-offs of different causal models. For instance, **DiscrepancyVAE (dVAE)** is primarily exploratory, aiming to discover latent variables and their causal graph from scratch (Zhang et al., 2023). In contrast, **SENA** takes a more targeted approach, acting as a confirmatory model for pathways while exploring perturbation effects; it **confirms** latent variables as known biological pathways and **explores** which pathway is targeted by a perturbation (de la Fuente et al., 2025).

The approach by de la Fuente et al. (2025) highlights the close relationship between SCM and CRL. By assuming its causal variables (the biological pathways) are known *a priori*, it operates like a traditional SCM. However, because it must still *learn* a representation of how to map raw gene expression data onto these variables and then discover the causal graph between them, it firmly resides within the CRL paradigm. It represents a specific, constrained form of CRL where prior knowledge heavily guides the representation learning process. This confirmatory design choice is motivated by the goal of enhancing the biological interpretability of the learned causal model.

However, this reliance on predefined pathways introduces significant limitations, primarily stemming from the challenge of knowledge transfer. Biological pathways exhibit strong cell-type and cell-line specificity, meaning that mechanisms characterized in one cellular context often do not generalize to another (Schneider et al., 2017; Gamazon et al., 2018; Hekselman & Yeger-Lotem, 2020). This context dependence poses a fundamental barrier to transferring pathway insights across different *in vitro* models and ultimately to patient tissues (Walter, 2019). Consequently, the findings from confirmatory models are difficult to apply to less-researched cell types, which limits the scalability and broader applicability of these approaches. Furthermore, this confirmatory strategy may not be suitable for smaller-scale experiments that only incorporate a limited set of genes, making it difficult to map observations to comprehensive pathway definitions.

## C  TRAINING CONFIGURATIONS AND ANALYSIS SETUP

### C.1  MODEL: ARCHITECTURES AND TRAINING HYPERPARAMETERS

To establish a fair and reproducible benchmark, we standardized key architectural and training hyperparameters for all models, primarily based on the default configurations provided in their respective publications and source code. This approach ensures that performance differences can be attributed to the models' core methodologies rather than variations in setup. Table S4 details the specific architectural parameters, such as latent dimension size and the number of transformer layers, while Table S5 outlines the training parameters used for each model, including optimizer, learning rate, and batch size. Notably, gradient clipping was employed exclusively for the scGPT model, as this is a standard and necessary technique to ensure numerical stability during the training of large transformer-based architectures.

For a complete list of hyperparameters and reproducibility instructions, please refer to the 'configs/' directory in our code repository and the accompanying 'README.md'.

Table S4: Detailed Model Architectures

| Hyperparameter | dVAE | SENA | scGPT | RAPTORGraph |
|---|---|---|---|---|
| Latent Dimension ($z_{dim}$) | 105 | 105 | N/A | 109 |
| Embedding Size ($embsize$) | N/A | N/A | 512 | N/A |
| Transformer Layers ($nlayers$) | N/A | N/A | 12 | N/A |
| Attention Heads ($nheads$) | N/A | N/A | 8 | N/A |
| Feed-Forward Dimension ($d_{hid}$) | N/A | N/A | 512 | 904 |
| Hidden Layers (Encoder) | 1 (128 units) | 1 (128 units) | N/A | N/A |
| Activation Functions | LeakyReLU | LeakyReLU | GELU | SiLU |

Table S5: Detailed Training Hyperparameters

| Hyperparameter | dVAE | SENA | scGPT | RAPTORGraph |
|---|---|---|---|---|
| Optimizer | Adam | Adam | AdamW | AdamW |
| Learning Rate | $1 \times 10^{-3}$ | $1 \times 10^{-3}$ | $1 \times 10^{-4}$ | $1 \times 10^{-4}$ |
| Training Batch Size | 32 | 32 | 16 | 256 |
| Evaluation Batch Size | 32 | 32 | 32 | 256 |
| Epochs | 100 | 100 | 30 | 100 |

# D EVALUATION METRICS AND LOSS FUNCTIONS

In this section, we provide detailed definitions for the key metrics used to quantitatively assess model performance. To ensure a comprehensive evaluation, we employed a suite of metrics targeting different aspects of model capability. We used the Mean Squared Error (MSE) for evaluating the reconstruction of basal cell states, the MK-MMD for evaluating the accuracy of out-of-distribution predictions for perturbed states, and a set of non-additive interaction scores to assess the model's ability to capture complex biological phenomena. The calculation of these metrics was standardized across all benchmarked tools.

## D.1 MEAN SQUARED ERROR

The MSE was used to measure each model's ability to faithfully reconstruct the gene expression profile of control (unperturbed) cells, with a lower MSE indicating better performance. To ensure an unbiased evaluation, the set of control cells was randomly split into two non-overlapping halves: an input set and a ground truth set. The evaluation proceeded in a batch-wise manner, where a batch of cells from the input set was fed to the model to generate reconstructions. The MSE was then calculated between these reconstructed cells and a corresponding batch from the ground truth set. The final reported MSE is the mean of the scores from all batches. For a single batch of $i$ cells with $n$ genes, the MSE is calculated as:

$$\mathcal{L}_{\text{MSE}} = \frac{1}{i \cdot n} \sum_{i=1}^{i} \sum_{j=1}^{n} (\mathbf{x}_{ij} - \hat{\mathbf{x}}_{ij})^2 \tag{S6}$$

where $\mathbf{x}_{ij}$ is the expression level of the $j$th gene in the $i$th true control cell, and $\hat{\mathbf{x}}_{ij}$ is the expression level of the $j$th gene in the $i$th reconstructed control cell.

## D.2 MAXIMUM MEAN DISCREPANCY

The MMD (Gretton et al., 2012) is a metric used to measure the distance between two probability distributions based on the distance between their mean embeddings in a Reproducing Kernel Hilbert Space (RKHS). It was used to measure the dissimilarity between the distribution of model-predicted gene expression profiles and the distribution of true, experimentally observed profiles for a given perturbation.

### D.2.1 THE KERNEL FUNCTION

The MMD relies on a kernel function, $k(\cdot, \cdot)$, which is a symmetric, positive definite function that computes a notion of similarity between two samples. A common choice for the kernel, and the one used in our work, is the Gaussian Gaussian Radial Basis Function (GRBF) kernel:

$$k(\mathbf{x}, \boldsymbol{y}) = \exp\left(-\frac{\|\mathbf{x} - \boldsymbol{y}\|^2}{2\sigma_{\text{MMD}}^2}\right) \tag{S7}$$

where $\sigma_{\text{MMD}}$ is the bandwidth hyperparameter.

### D.2.2 FORMAL DEFINITIONS OF MMD

The squared MMD between the true data distribution $p_{\text{data}}$ and the model's learned distribution $p_{\text{model}}$ is defined in its population form as:

$$\begin{aligned}
\text{MMD}^2(p_{\text{data}}, p_{\text{model}}) = \; & \mathbb{E}_{\mathbf{x}, \mathbf{x}' \sim p_{\text{data}}}\big[k(\mathbf{x}, \mathbf{x}')\big] \\
& - 2\mathbb{E}_{\mathbf{x} \sim p_{\text{data}}, \hat{\mathbf{x}} \sim p_{\text{model}}}\big[k(\mathbf{x}, \hat{\mathbf{x}})\big] \\
& + \mathbb{E}_{\hat{\mathbf{x}}, \hat{\mathbf{x}}' \sim p_{\text{model}}}\big[k(\hat{\mathbf{x}}, \hat{\mathbf{x}}')\big]
\end{aligned} \tag{S8}$$

In practice, we use batches of data and compute the unbiased empirical estimator. Given a batch of $i$ true samples $\{\mathbf{x}_i\}_{i=1}^{i} \sim p_{\text{data}}$ from matrix $\boldsymbol{X}$ and a batch of $j$ predicted samples $\{\hat{\mathbf{x}}_j\}_{j=1}^{j} \sim p_{\text{model}}$

from matrix $\hat{\boldsymbol{X}}$, the estimator is:

$$\text{MMD}_u^2(\boldsymbol{X}, \hat{\boldsymbol{X}}) = \frac{1}{i(i-1)} \sum_{i \neq j}^{i} k(\mathbf{x}_i, \mathbf{x}_j)$$
$$+ \frac{1}{j(j-1)} \sum_{i \neq j}^{j} k(\hat{\mathbf{x}}_i, \hat{\mathbf{x}}_j)$$
$$- \frac{2}{ij} \sum_{i=1}^{i} \sum_{j=1}^{j} k(\mathbf{x}_i, \hat{\mathbf{x}}_j) \qquad \text{(S9)}$$

### D.2.3  MULTI-KERNEL MAXIMUM MEAN DISCREPANCY

MK-MMD (Ren et al., 2010; Zhu et al., 2017) extends the standard MMD by using a convex combination of $L$ different kernels (Gretton et al., 2012). The squared MK-MMD is defined as a weighted sum of individual squared MMDs:

$$\text{MK-MMD}^2(p_{\text{data}}, p_{\text{model}}) = \sum_{l=1}^{L} \beta_l \cdot \text{MMD}_{k_l}^2(p_{\text{data}}, p_{\text{model}}) \qquad \text{(S10)}$$

where $\beta_l \geq 0$ are the weights assigned to each kernel.

### D.2.4  RATIONALE FOR USING MK-MMD

The choice of MK-MMD over MSE for evaluating interventional predictions is motivated by two key factors. First, MK-MMD captures holistic distributional shifts, including changes in variance, skewness, and modality, whereas MSE is only sensitive to the mean. This is critical for modeling the heterogeneous biological response to a perturbation. Second, state-of-the-art CRL models for interventional data often do not produce one-to-one pairings between predicted and ground-truth samples, making MSE computation impossible. MMD is a two-sample test that compares two sets of samples directly, which aligns perfectly with this setting.

### D.2.5  HYPERPARAMETER SELECTION FOR MK-MMD

The performance of the MMD test is highly sensitive to the kernel bandwidth, $\sigma_{\text{MMD}}$. To establish a fair and consistent benchmark across all models, it was necessary to standardize this parameter. We noted that different models, such as DiscrepancyVariational Autoencoder (VAE) and SENA (de la Fuente et al., 2025), use different default values. The implementations often define a fix_sigma parameter, which relates to the standard kernel variance by fix_sigma = $2\sigma_{\text{MMD}}^2$. To standardize, we aligned with the value used in SENA. Our benchmark, therefore, uses a MK-MMD implementation with a base fix_sigma set to 200.0 for all evaluations, which corresponds to a variance $\sigma_{\text{MMD}}^2$ of 100.0. This approach generates a series of kernels with varying bandwidths derived from this base value, ensuring the metric's scale and sensitivity were consistent and robust.

### D.3  BENCHMARKING ON REPLOGLE2020 DATASET

First, we measured the MSE between the ground truth and predicted profiles of control cells to evaluate how well each model reconstructs the basal, unperturbed cellular state. Second, we calculated the MK-MMD between the distributions of ground truth and predicted profiles for double-gene perturbations to assess the model's ability to capture the global transcriptional shift following a complex intervention.

| Metric | scGPT | RAPTORGraph (ours) |
|--------|-------|--------------------|
| MK-MMD | $0.138 \pm 0.000$ | $\mathbf{0.137 \pm 0.000}$ |
| MSE | $0.069 \pm 0.000$ | $\mathbf{0.054 \pm 0.000}$ |

Table S6: Results using the Replogle2020 dataset are averaged over 3 runs. Values are mean $\pm$ variance.

To validate the generalization capabilities of RAPTORGraph across different experimental contexts and cell lines, we extended our evaluation to include the large-scale single-cell CRISPR screen dataset from Replogle et al. (2020) as shown in Table S6. As discussed in Sec. 5, our primary comparative analysis prioritized the Norman et al. (2019) dataset due to the rigid architectural dependency of several baseline models (specifically SENA and Discrepancy VAE) on auxiliary files and preprocessing pipelines unique to that study. Adapting these confirmatory models to the distinct structure of the Replogle2020 dataset proved non-trivial. Additionally, we excluded the model because it behaves in a fundamentally different way that is incompatible with our evaluation framework, making a fair and direct comparison impossible. Consequently, this supplementary benchmark is restricted to a direct comparison between RAPTORGraph and , demonstrating our model's robust performance and superior reconstruction fidelity on independent biological data.

## D.4 NON-ADDITIVE GENETIC INTERACTION ANALYSIS

To evaluate the models' ability to predict non-additive effects, we designed a genetic interaction analysis inspired by the Precision@10 metric introduced in the GEARS paper (Roohani et al., 2022). While the original study used a robust regression model, we opted for a more direct naive additive baseline, defined as the vector sum of single-perturbation effects over control ($\Delta_A + \Delta_B$). We calculated five interaction scores by comparing the predicted double-perturbation effect ($\Delta_{AB}$) to this baseline. This unified metric was applied to all models to ensure a fair comparison.

The analysis follows a multi-step process for each double-gene perturbation. First, "effect vectors" ($\Delta$) are calculated for single and double perturbations by subtracting the mean gene expression profile of control cells from the mean profile of perturbed cells. This is done for both ground truth data and model predictions. The naive additive baseline assumes the double perturbation effect is a linear sum of individual effects:

$$\Delta(A + B)_{\text{naive}} = \Delta A + \Delta B \tag{S11}$$

By comparing the true or predicted $\Delta(A + B)$ to the individual and naive additive vectors, we compute raw scores for five distinct genetic interaction subtypes.

- *Synergy* and *Suppression* are measured using a ratio of magnitudes:

$$\text{Synergy Ratio} = \frac{\|\Delta(A + B)\|_2}{\|\Delta(A + B)_{\text{naive}}\|_2} \tag{S12}$$

- *Neomorphism* measures changes in the direction of the effect:

$$\text{Neomorphism Score} = 1 - \cos\left(\Delta(A + B), \Delta(A + B)_{\text{naive}}\right) \tag{S13}$$

- *Redundancy* measures functional overlap:

$$\text{Redundancy Score} = \min\left(\cos\left(\Delta(A + B), \Delta A\right), \cos\left(\Delta(A + B), \Delta B\right)\right) \tag{S14}$$

- *Epistasis* measures dominance:

$$\text{Epistasis Score} = \left|\cos\left(\Delta(A + B), \Delta A\right) - \cos\left(\Delta(A + B), \Delta B\right)\right| \tag{S15}$$

Final performance is measured using Precision@10. For each interaction type, perturbations are ranked by their predicted score, and the top 10 are selected. These are compared against a ground truth set of interactors, defined as those whose true score falls in the top 25% (or bottom 25% for suppression) for that category. The Precision@10 score is the fraction of correct predictions in the top 10.

## D.5 REVERSE PERTURBATION ANALYSIS

To assess the models' understanding of genotype-phenotype relationships, we implemented an in silico reverse perturbation prediction task, inspired by the analysis in the scGPT study (Cui et al., 2024). This task challenges a model to infer the causal genetic perturbation that induced a given transcriptomic state.

The analysis is conducted on a specific subset of 20 genes from the Norman et al. (2019) dataset, creating a controlled combinatorial search space. For each unique perturbation condition $p$, we

establish a ground truth mean expression profile, $\bar{\mathbf{x}}_p$, by averaging the expression vectors of all cells for that condition:

$$\bar{\mathbf{x}}_p = \frac{1}{N_p} \sum_{i=1}^{N_p} \mathbf{x}_{p,i} \tag{S16}$$

where $N_p$ is the number of cells under perturbation $p$, and $\mathbf{x}_{p,i} \in \mathbb{R}^n$ is the expression profile of the $i$-th cell. The central component is the creation of a comprehensive in silico prediction database restricted to the 20-gene subset. For each model, we generate a predicted expression profile $\hat{\mathbf{x}}_{A+B}$ for every unique combination of two genes, A and B. This is achieved by feeding the model the ground truth mean control profile, $\bar{\mathbf{x}}_{\text{ctrl}}$, and conditioning it on the desired double perturbation. The prediction function $f$ for a given model is expressed as:

$$\hat{\mathbf{x}}_{A+B} = f(\bar{\mathbf{x}}_{\text{ctrl}}, \text{pert} = A + B) \tag{S17}$$

This results in a key-value database where each key is a double-perturbation identifier from the 20-gene subset, and the value is the corresponding predicted mean expression vector in $\mathbb{R}^n$.

With the prediction database established, we query it using the ground truth mean expression profiles of the experimentally observed double perturbations from the 20-gene subset. For each true double perturbation $p_{true}$ present in the test set, its ground truth mean profile $\bar{\mathbf{x}}_{p_{\text{true}}}$ is used as a query vector. We then calculate the Euclidean distance between this query vector and every predicted vector $\hat{\mathbf{x}}_{p_{\text{pred}}}$ in the database:

$$d(\bar{\mathbf{x}}_{p_{\text{true}}}, \hat{\mathbf{x}}_{p_{\text{pred}}}) = \sqrt{\sum_{j=1}^{n} (\bar{\mathbf{x}}_{p_{\text{true}}, j} - \hat{\mathbf{x}}_{p_{\text{pred}}, j})^2} \tag{S18}$$

The perturbations in the database are subsequently ranked based on their proximity to the query vector, from the smallest to the largest Euclidean distance. This yields a ranked list of predicted perturbations that are most likely to have caused the observed cellular state.

To evaluate the ranking performance, we employ the "Hit Rate @ K" metric, which measures the frequency of successful retrievals within the top K predictions. We define two variants of this metric to capture different aspects of prediction accuracy.

*Correct Hit Rate @ K*: This is a stringent metric that measures the proportion of queries where the exact true perturbation is correctly identified within the top K ranked predictions. It is formally defined as:

$$\text{Correct Hit Rate @ K} = \frac{1}{|\mathcal{Q}|} \sum_{q \in \mathcal{Q}} \mathbb{I}(\text{rank}(p_q) \leq K) \tag{S19}$$

where $\mathcal{Q}$ is the set of all query perturbations, $p_q$ is the true perturbation corresponding to query $q$, and $\mathbb{I}$ is the indicator function, which is 1 if the condition is met and 0 otherwise.

*Relevant Hit Rate @ K*: This is a more lenient metric that assesses whether the model can identify at least one of the correct genetic components of a combinatorial perturbation. A prediction is considered "relevant" if its set of perturbed genes has a non-empty intersection with the set of genes in the true perturbation. The metric is defined as:

$$\text{Relevant Hit Rate @ K} = \frac{1}{|\mathcal{Q}|} \sum_{q \in \mathcal{Q}} \mathbb{I}(\exists \hat{p} \in \text{TopK}(q) \text{ s.t. genes}(\hat{p}) \cap \text{genes}(p_q) \neq \emptyset) \tag{S20}$$

where $\text{TopK}(q)$ is the set of top K predicted perturbations for query $q$, and $\text{genes}(p)$ is the set of constituent genes in perturbation $p$. This metric rewards models that can correctly identify influential genes, even if the exact combination is not perfectly predicted. Together, these two metrics provide a comprehensive evaluation of each model's ability to reverse-engineer the mapping from cellular phenotype back to genetic cause.

## D.6 GENE SET ENRICHMENT ANALYSIS

To validate the semantic meaning of the learned Directed Acyclic Graph, we performed a systematic Gene Set Enrichment Analysis. We employed an in-silico perturbation strategy to define the biological identity of each latent variable:

1. Latent Perturbation: For each causal latent factor $\mathbf{z}_i$ associated with a known gene perturbation, we artificially activated the factor in the latent space by setting the condition vector $\mathbf{c}$ such that the target component $\mathbf{c}_i = 1$ and all other components $\mathbf{c}_{j \neq i} = 0$.

2. Counterfactual Decoding: We decoded this perturbed latent state back to the gene expression space $\mathbb{R}^n$ to generate a batch of "counterfactual" expression profiles.

3. Differential Analysis: We computed the mean differential expression vector $\delta = \bar{x}_{int} - \bar{x}_{ctrl}$ relative to the control baseline.

4. Enrichment This vector was used to create a ranked gene list, which was analyzed using the `gseapy` framework against the MSigDB Hallmark 2020 gene set collection.

This process allows us to independently verify if the latent factor $\mathbf{z}_i$ actually encodes the biological function of its assigned gene $g_i$, assigning a verifiable "biological label" to the nodes of our learned graph.

### D.7 RATIONALE FOR THE EXCLUSION OF GEARS FROM THE BENCHMARK

In designing this benchmark, our goal was to establish a fair and methodologically consistent framework for comparing models that predict the full distribution of single-cell gene expression profiles following perturbation. While we considered including other prominent models such as GEARS, we ultimately excluded it from the final comparison due to fundamental incompatibilities between its predictive paradigm and our standardized evaluation framework. The two primary reasons for its exclusion are its prediction of a single population-average vector rather than a cell distribution, which is incompatible with our MMD metric, and its non-equivalent handling of the control state, where it uses the precomputed mean of true control cells as a baseline rather than learning to reconstruct them. Given these significant differences, we concluded that a fair and direct comparison between GEARS and the other models within our established framework was not possible.

# E MODEL ARCHITECTURE

This section provides a comprehensive overview of the core architectural components of our model, detailing both the novel GraphPathway Encoder and the deep causal graph learning framework.

## E.1 OVERVIEW OF RAPTORGRAPH FRAMEWORK

The architecture of the **R**esponse **A**nalysis of **P**erturbed **T**ranscript**O**mes using Inte**R**pretable **Graph** (RAPTORGraph) framework, illustrated in Fig. S4, is designed as an end-to-end system for learning causal relationships from interventional single-cell data. The framework is composed of three primary modules: a preconditioned encoder for sparse representation learning, a DAG discovery layer for identifying causal structure, and a decoder for reconstructing cellular states.

The input to the model is the high-dimensional gene expression data. The first layer of the encoder is our novel preconditioned GraphPathway Encoder, which addresses the Intervention Spillover Problem. By leveraging prior biological knowledge, it enforces a sparse mapping from the thousands of observed genes ($\mathbb{R}^n$) to a lower-dimensional set of learned meta-pathways ($\mathbb{R}^d$). This ensures that a single-gene intervention activates only a sparse, well-defined set of learned meta-pathways in the latent space, fulfilling the requirements for valid downstream causal inference.

The resulting latent representations, $\mathbf{z}$, are then passed to the DAGMA-based DAG discovery module. This layer operates directly on the latent variables to learn the causal relationships *between* the learned meta-pathways. Its primary output is a directed acyclic graph ($\mathcal{G}$), represented by its adjacency matrix, which models the flow of biological influence among the learned meta-pathways.

Finally, the decoder network reconstructs the original gene expression profile from the latent representation. This ensures that the latent variables, in addition to being causally structured, also retain sufficient information to accurately capture the full cellular state. The model is trained end-to-end by optimizing a composite loss function that balances these different objectives, as detailed in the figure caption.

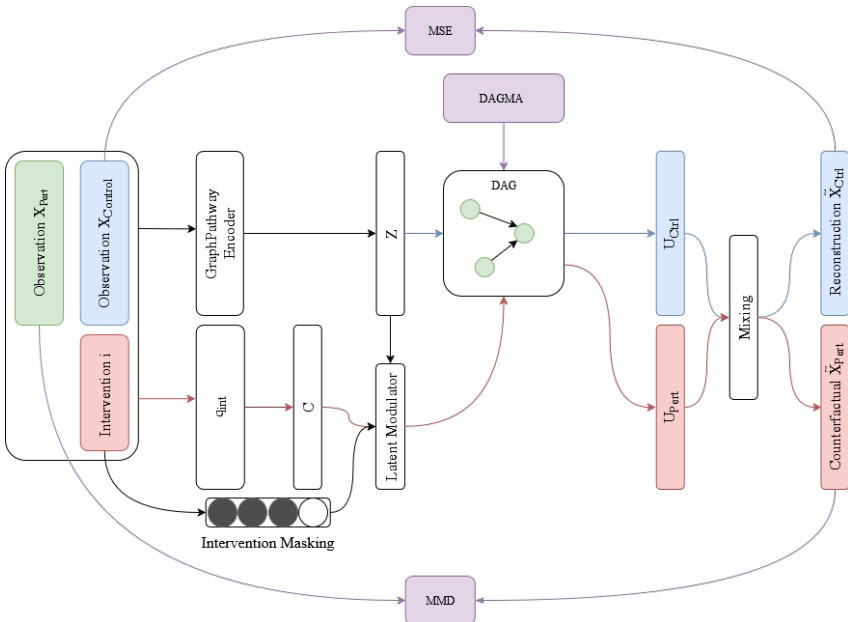

Figure S4: Detailed schematic of the RAPTORGraph framework. The preconditioned GraphPathway Encoder enforces a sparse mapping from the high-dimensional gene expression space ($\mathbb{R}^n$) to the learned meta-pathways ($\mathbb{R}^d$). The DAGMA-based DAG operates on the latent representations to learn a directed acyclic graph ($\mathcal{G}$) representing the causal relationships between the learned meta-pathways. The model is trained end-to-end using a composite loss function comprised of several terms: a reconstruction loss (MSE), a distributional loss (MK-MMD) on interventional predictions, a KL Divergence on the VAE latent variables, and an acyclicity constraint from the DAGMA layer.

### E.2 GRAPHPATHWAY ENCODER ARCHITECTURE

The GraphPathway Encoder is a custom neural network layer designed to solve the Intervention Spillover Problem through a principled architectural design. It combines a preconditioned weight matrix to deconfound known perturbation effects with a multi-subgraph structure to capture complex, non-linear gene interactions.

#### E.2.1 DECONFOUNDING THROUGH PRECONDITIONING

The foundational component of the encoder is its preconditioned weight matrix, which enforces the single-node intervention assumption required for causal identifiability. As described in Sec. 3.1, the layer's primary weight matrix $\boldsymbol{W}$ is structured as a block matrix with a strictly diagonal top-left block (Eq. (4)). This structure guarantees that each known perturbed gene has a direct, isolated influence on exactly one latent pathway, while the zero-block explicitly prevents its effect from spilling over into other pathways. This design transforms the encoder into a deterministic modulator for known interventions, allowing the intervention mask $\boldsymbol{m}$ to be fixed by the architecture rather than being a learned (and potentially dense) variable. By explicitly blocking the spillover of the intervention effect into other pathways via the zero-block, this design transforms the encoder into a deterministic modulator for known interventions, allowing the intervention mask $\boldsymbol{m}$ to be fixed by the architecture rather than being a learned (and potentially confounded) variable.

#### E.2.2 MODELING NON-LINEAR INTERACTIONS WITH SUBGRAPHS

To move beyond a purely linear model and capture the complex, non-linear dynamics of biological processes, the architecture learns $m$ distinct representations for each pathway. This is achieved by first applying a masked linear layer that maps the input genes $\mathbf{x}$ to an intermediate representation. This layer's weight matrix is derived by repeating the base preconditioned mask $m$ times, creating $m$ parallel, similarly conditioned subgraphs that are initialized differently to learn diverse feature mappings. The initial activation vector, $\mathbf{s}^{(g)}$, for each subgraph $g$ is computed as:

$$\mathbf{s}^{(g)} = \rho(\boldsymbol{W}^{(1,g)}\mathbf{x} + \mathbf{b}^{(1,g)}) \quad \text{for } g = 1, \ldots, m \tag{S21}$$

where each layer-specific weight matrix $\boldsymbol{W}^{(1,g)}$ has the same block-diagonal structure defined in Eq. (4).

An intermediate *Interaction Block* then processes these subgraph activations to model higher-order dependencies. This block applies a series of pathway-specific linear transformations and non-linear activations to the set of $m$ activations within each pathway, enabling the model to learn complex interactions between the different feature mappings before the final aggregation step.

Finally, the processed subgraph activations are aggregated into the final pathway representations using a grouped 1D convolution. This operation has a kernel size of $m$ and is configured with $d$ groups, making it equivalent to applying a separate, pathway-specific linear layer to the set of $m$ subgraph activations for each pathway. This learns an optimal linear combination of the non-linear subgraph features to produce the final parameters for the latent distributions, $\mu_{\mathbf{z}}$ and $\sigma_{\mathbf{z}}$.

#### E.2.3 IMPLEMENTATION DETAILS OF GRAPHPATHWAY

The specific implementation used in our model is the 'InteractingGraphPathwayLayer'. This layer encapsulates the preconditioning, multi-subgraph mapping, and interaction blocks into a single module. To model complex, non-linear biological processes, each pathway learns its own unique set of interaction weights within the interaction block. This design provides greater expressive power, as it allows the model to capture distinct interaction dynamics for each biological process. When configured for a variational autoencoder, the output channel dimension of the final grouped convolution is doubled, allowing the layer to directly produce both the mean $\mu_{\mathbf{z}}$ and variance $\sigma_{\mathbf{z}}$ for each latent pathway.

### E.3 Deep Causal Graph Learning with DAGMA

This subsection details the DAGMA framework for causal discovery, the rationale for its adaptation within our deep learning model, and the key implementation principles required for stable and effective graph learning in a VAE context.

#### E.3.1 Theoretical Background: From NOTEARS to DAGMA

A fundamental challenge in causal discovery is the combinatorial nature of learning a DAG, as the number of possible graphs grows super-exponentially with the number of variables. A breakthrough was achieved by NOTEARS (Zheng et al., 2018), which reformulated this discrete problem into a continuous optimization problem amenable to gradient-based methods. The core innovation was a differentiable function, $h(\boldsymbol{A})$, whose value is zero if and only if the weighted adjacency matrix $\boldsymbol{A}$ represents a DAG. The original NOTEARS constraint is based on the matrix exponential:

$$h(\boldsymbol{A}) = \text{Tr}(e^{\boldsymbol{A} \circ \boldsymbol{A}}) - d = 0 \tag{S22}$$

where $\text{Tr}(\cdot)$ is the trace of the matrix and $\boldsymbol{A} \circ \boldsymbol{A}$ is the element-wise (Hadamard) product. This function cleverly sums all cycles of all possible lengths in the graph; it equals zero only if no cycles exist.

DAGMA (Bello et al., 2022) builds upon this continuous optimization framework but introduces a new, log-determinant-based acyclicity characterization that often leads to faster and more stable convergence. The DAGMA acyclicity constraint is:

$$h_s(\boldsymbol{A}) = -\log \det(s\boldsymbol{I} - \boldsymbol{A} \circ \boldsymbol{A}) + d \log s = 0 \tag{S23}$$

where $s > 0$ is a scaling scalar. This function is based on the principle that a matrix corresponds to a DAG if and only if all eigenvalues of its squared adjacency matrix are zero. The log-determinant acts as a barrier, approaching infinity as the graph's structure approaches a cycle, while being minimized at zero exclusively for DAGs.

#### E.3.2 The DAGMA Optimization Framework

The full optimization problem is to minimize a score function that measures how well the graph fits the data, subject to the acyclicity constraint. For a linear model with latent variables $\mathbf{z}$, the score function $S(\boldsymbol{A})$ is typically the mean squared error of reconstruction:

$$S(\boldsymbol{A}) = \frac{1}{2n} \|\mathbf{z} - \mathbf{z}\boldsymbol{A}\|_F^2 \tag{S24}$$

The original DAGMA paper proposes a path-following approach that solves a sequence of unconstrained optimization problems:

$$\min_{\boldsymbol{A}} (\mu \cdot S(\boldsymbol{A}) + h_s(\boldsymbol{A})) \tag{S25}$$

where $\mu$ is a hyperparameter that is gradually decreased towards zero during training. As $\mu \to 0$, the penalty on violating acyclicity becomes infinitely strong, forcing the final solution for $\boldsymbol{A}$ to be a DAG.

#### E.3.3 Rationale for a Modified DAGMA in a Causal Representation Learning

Integrating DAGMA into a VAE for interventional CRL requires careful design to ensure the model learns a meaningful causal graph. Our implementation is guided by two core principles.

**Principle 1: Learning the Invariant Graph from Observational Data.** A foundational assumption is that the causal graph $\boldsymbol{A}$ represents an invariant, underlying mechanism of the system. Therefore, the DAGMA loss, which encourages the learning of this structure, must be computed exclusively on the latent representations of observational data ($\mathbf{z}_{\text{obs}}$). This teaches the model the fundamental causal rules from the system's natural dynamics. The interventional data and their latent representations ($\mathbf{z}_{\text{int}}$) are used to compute a separate discrepancy loss (e.g., MMD). This second loss teaches the model the consequences of breaking the causal rules, evaluating how well the learned graph functions as a simulator for interventions. Applying the graph loss to interventional latents would create a conflicting objective, as an intervention by definition breaks the natural mechanism the graph is supposed to represent.

**Principle 2: Isolating Gradient Flow via Input Detachment.** The most critical design principle for stability is isolating the gradient flow between the VAE's encoder and the DAGMA layer. This is achieved by detaching the latent variable tensor ($\mathbf{z}_{\text{obs}}$) before it is used in the score calculation ($S(\boldsymbol{A}, \mathbf{z}_{\text{obs}}.\text{detach}())$). This design choice has two key benefits. First, it mimics the original DAGMA algorithm, which operates on a fixed dataset. Second, and more importantly, it creates a separation of concerns: the encoder learns to produce a rich representation, and the DAGMA layer learns the graph of that representation. Without detaching, gradients from the DAGMA score would flow back to the encoder, encouraging it to learn a trivial, simplistic latent space that is easy to fit, thereby destroying the quality of the representation.

### E.3.4 IMPLEMENTATION DETAIL OF DAGMA

Several practical considerations are crucial for the stable application of DAGMA in a deep learning context.

**Weight Initialization and Invertibility.** To ensure stability from the start of training, the weights of the adjacency matrix $\boldsymbol{A}$ are initialized from a uniform distribution in the range $[-0.1, 0.1]$. This provides the optimizer with small, non-zero weights as a starting point. This initialization also makes it highly probable that the matrix $(\boldsymbol{I} - \boldsymbol{A})$ is well-conditioned and invertible, which is critical for the forward pass of the linear DAGMA model. The forward pass solves the structural equation $\mathbf{z} = \mathbf{z}\boldsymbol{A} + \mathbf{z}$ for $\mathbf{z}$, which requires computing $\mathbf{z}(\boldsymbol{I} - \boldsymbol{A})^{-1}$. To handle potential numerical instability, our implementation first attempts to solve the linear system directly. If this fails due to a singular or ill-conditioned matrix, it robustly falls back to using the Moore-Penrose pseudo-inverse, guaranteeing a stable forward pass throughout training.

**Input Normalization.** A common failure mode is the collapse of the graph weights in $\boldsymbol{A}$ to a trivial near-zero solution. This is prevented by applying a normalization layer (e.g., 'LayerNorm') to the latent inputs ($\mathbf{z}_{\text{obs}}$ and $\mathbf{z}_{\text{int}}$) before they enter the DAGMA layer. This ensures the score term has a consistent magnitude, providing a strong and stable gradient signal that encourages the model to learn a non-trivial graph.

**Numerical Precision.** The acyclicity constraint $h_s(\boldsymbol{A})$ is theoretically guaranteed to be non-negative. However, due to floating-point limitations, it can sometimes become a very small negative number. To handle this, the calculated value is clamped at zero, which respects the theoretical constraint while gracefully handling practical numerical artifacts.

### E.4 RATIONALE FOR GLOBAL PAIRING THROUGH OPTIMAL TRANSPORT

A primary objective in the field of single-cell genomics is to predict the impact of genetic modifications (e.g., via CRISPR) on a cell's gene expression profile. Data for this task usually comprises two distinct, unpaired populations of cells: an observational control group and an interventional perturbed group.

Crucially, there is no direct one-to-one mapping between these populations. A cell from the perturbed group and a cell from the control group are distinct biological entities. It is impossible to ascertain what a perturbed cell would have looked like prior to the intervention, which makes learning the treatment effect a fundamental counterfactual problem. Therefore, the central challenge is to identify a general transformation that maps the control distribution to the perturbed distribution in the absence of paired examples.

To address this counterfactual gap, models such as DiscrepancyVAE and SENA often use a simplified approach of creating training pairs by randomly selecting an observational (control) sample from the same mini-batch to match an interventional sample. While this random pairing is computationally efficient, it is a primary cause of mean collapse.

For a given interventional sample (e.g., cell $i$, in which gene $A$ was perturbed), the randomly selected observational sample (e.g., cell $j$) is highly unlikely to be an appropriate counterfactual. The biological state of cell $j$ may differ substantially from that of cell $i$ prior to the perturbation. Consequently, the model must learn a transformation from an often noisy and biologically irrelevant initial state. Across thousands of training iterations, a single interventional sample is paired with

hundreds of different random controls. Each pairing provides a distinct and noisy gradient signal. The optimiser seeks to minimise the loss on average across these inconsistent pairings and converges on a simple solution: predicting the mean of the target distribution. The random noise from the poor counterfactuals averages to zero, leaving only the central tendency as a stable learning signal. Ultimately, the objective function effectively becomes an expectation over these random pairings. The model learns that the most reliable strategy to achieve a low expected loss is to ignore the specifics of the randomly chosen input and produce an average output. The optimisation landscape is thereby smoothed in a manner that makes the 'mean prediction' an attractive local minimum.

Rather than creating noisy one-to-one random pairs, optimal transport (OT) considers the entire population of observational and interventional latents. It computes an optimal flow of probability mass from the observational to the interventional distribution. This flow represents the most efficient method of morphing one entire point cloud into the other. The flow implicitly creates meaningful pairings. Each interventional sample is matched with the observational sample(s) that are closest in gene expression space, providing the best possible counterfactuals within the given populations. The theoretical properties of OT ensure that it robustly preserves the inherent heterogeneous subpopulation structures by minimizing the ground movement cost, a feature critical for reliable counterfactual prediction.

### E.5 Training and Test Data Partitioning

To create a robust benchmark for evaluating the generalization capabilities of each model, we partitioned the Norman et al. (2019) dataset based on the type of perturbation. The training dataset was composed of the complete set of control cells (i.e., those with non-targeting guides) and the complete set of cells from all available **single-gene perturbation** conditions. This approach ensures that the models are trained on the fundamental effects of individual gene knockdowns. The test dataset, conversely, consisted exclusively of cells from all **double-gene perturbation** conditions. These combinatorial perturbations were entirely held out from the training process. This training/test split was designed to specifically assess each model's ability to perform out-of-distribution prediction, where the primary task is to predict the effects of unseen combinatorial perturbations based on the learned effects of their individual constituent perturbations.

# F EXTENDED RESULTS

## F.1 EXTENDED RESULTS: REVERSE PERTURBATION ANALYSIS

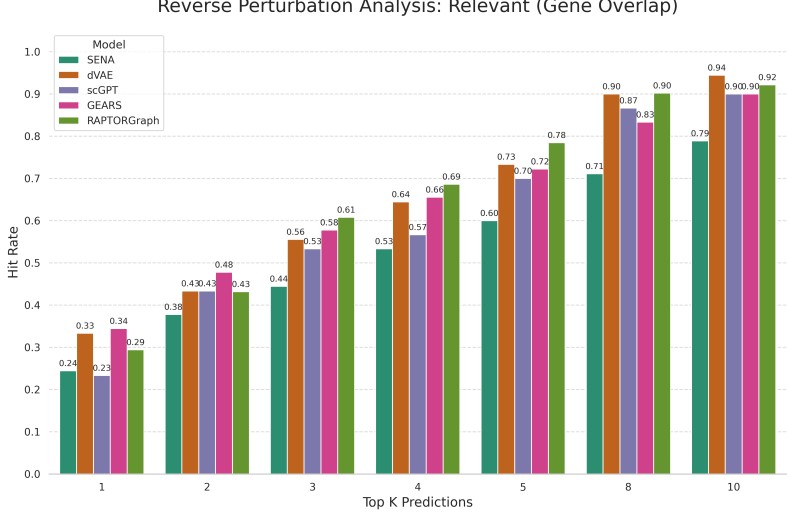

Figure S5: Hit Rate @ K (correct) for the reverse perturbation task. The metric queries where the exact true perturbation is correctly identified within the top K ranked predictions. Results are averaged over 3 runs.

Figure S6: Hit Rate @ K (relevant) for the reverse perturbation task. The metric measures the fraction of queries where at least one correct gene was identified in the top K predictions. Higher values are better. Results are averaged over 3 runs.

## F.2 EXTENDED RESULTS: BIOLOGICAL VALIDATION OF LEARNED META-PATHWAYS

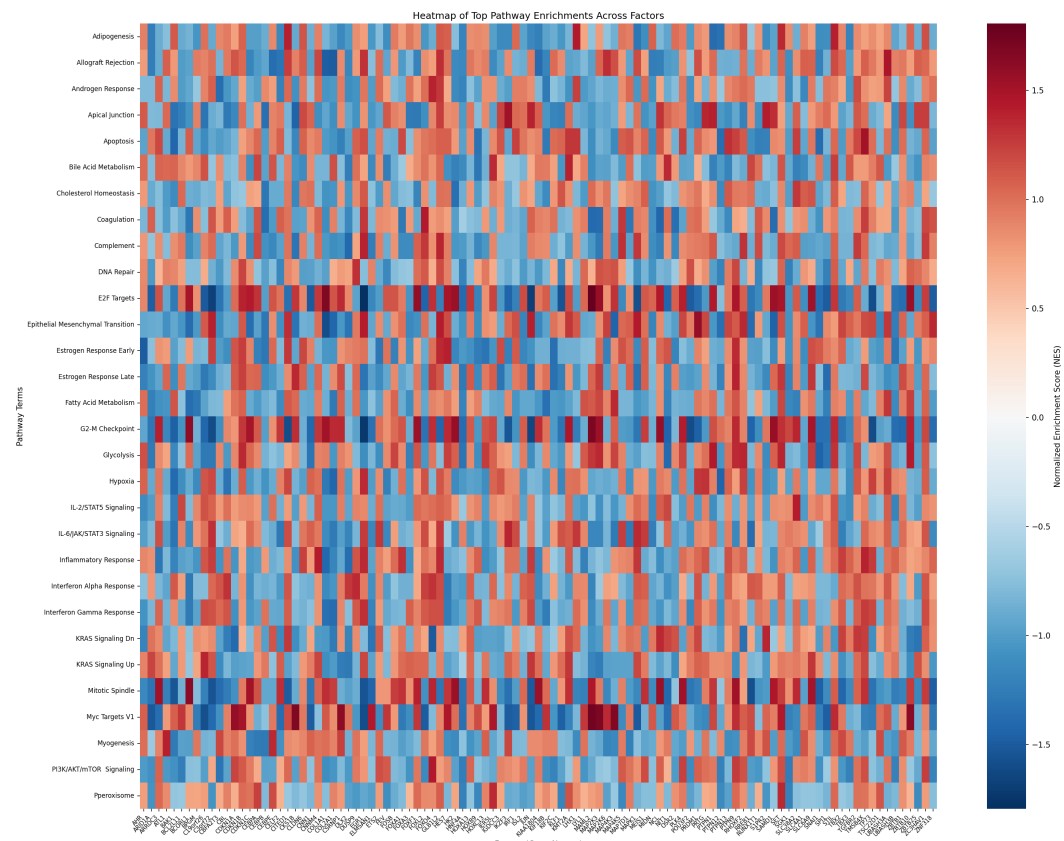

Figure S7: Heatmap of Top Pathway Enrichments Across Learned Latent Factors. Each column corresponds to a specific latent factor ($z_i$) in the RAPTORGraph model. Importantly, these labels are not post-hoc assignments; due to the GraphPathway encoder's preconditioning, each latent factor $z_i$ is architecturally constrained to mediate the effect of a specific gene perturbation (labeled on the x-axis). The rows represent biological pathways from the MSigDB Hallmark collection. The color intensity indicates the Normalized Enrichment Score (NES) derived from in-silico perturbation of that factor. The strong concordance between the architectural label and the biological result confirms that the model has successfully encoded the specific biological function of each gene into its corresponding latent variable.

The GSEA analysis reveals that RAPTORGraph captures distinct and biologically accurate programs. Figure S7 visualizes the Normalized Enrichment Scores (NES) for the top enriched pathways across the learned factors. We highlight distinct biological programs captured by the model, verified against the known functions of their corresponding gene perturbations:

1. **Differentiation-Induced Arrest:** The latent factor linked to *EGR1* displays massive negative enrichment for "E2F Targets" (NES $-1.79$, FDR $\approx 0.0$) and "G2-M Checkpoint" (NES $-1.72$, FDR $\approx 0.0$). EGR1 is a known tumor suppressor in K562 leukemia cells that drives differentiation along the megakaryocytic lineage, a process that necessitates a complete exit from the cell cycle (Ma et al., 2019). RAPTORGraph correctly identifies this potent anti-proliferative program.

2. **Stress-Induced Growth Arrest:** The latent factor linked to the AP-1 subunit *JUN* shows strong negative enrichment for "E2F Targets" (NES $-1.69$, FDR $< 0.001$) and "G2-M Checkpoint" (NES $-1.63$, FDR $< 0.002$). While often associated with inflammation, c-Jun is a central stress-response mediator; in the context of K562 cells, its activation drives a termination of cell division required for stress adaptation or differentiation (Shaulian & Karin, 2002; Shaulian et al., 2000).

3. **p53-Like Tumor Suppression:** The latent factor associated with *TP73* displays significant negative enrichment for "G2-M Checkpoint" (NES $-1.60$, FDR $< 0.03$). *TP73* is a functional homolog of *TP53* (p53); its activation triggers downstream checkpoint signaling that arrests the cell cycle to prevent genomic instability (Kaghad et al., 1997).

These findings confirm that the nodes in our learned DAG represent specific, identifiable biological mechanisms. Consequently, the edges learned by the DAGMA module can be interpreted as causal regulatory links between these verified biological processes.

### F.3 EXTENDED RESULTS: COMPUTATIONAL COMPLEXITY OF OPTIMAL TRANSPORT PREPROCESSING

We evaluated the computational cost of the proposed OT pairing strategy compared to a random baseline. While random pairing scales linearly ($O(N)$), the OT approach involves solving the Earth Mover's Distance (EMD) problem, which typically exhibits a worst-case complexity of $O(N^3 \log N)$.

To mitigate this, the OT pairing problem is formulated by taking the entire control population ($N_{ctrl} = 8,907$) and pairing it independently with each specific intervened population (i.e., cells sharing the same perturbation label, $N_k$ in the hundreds). This effectively segments the total perturbed dataset ($N = 108,497$) into many smaller, manageable OT problems, significantly reducing the $N$ in the $O(N^3 \log N)$ complexity for each individual transport plan and enabling efficient parallelization.

We quantified the overhead of OT preprocessing on a workstation equipped with an Intel Core i9-9900K CPU (8 cores, 16 threads) and 64 GB RAM. The OT preprocessing added approximately 34 seconds to the baseline execution time (Random: 1m 44s vs. OT: 2m 18s). Detailed profiling confirmed that the process is compute-bound but highly efficient:

- **Parallelization**: The process achieved a peak CPU utilization of 1140%, confirming that the the strategy effectively saturates available cores ($\approx 11.4$ active threads).
- **Memory**: Peak memory usage was modest at 2.4 GB.

To contextualize this cost, we benchmarked the total training time for a standard 100-epoch experiment on an NVIDIA RTX 3090 GPU ($\approx 25.8$ minutes). The 34-second OT overhead represents only 2.2% of the total experimental runtime. Given that OT is critical for preserving causal identifiability and preventing mean collapse, this negligible computational cost presents a highly favorable trade-off.

## F.4 Extended Results: Empirical Analysis of Loss Functions for Sparse Single-Cell Causal Inference

Training causal inference models on single-cell RNA sequencing (scRNA-seq) data presents unique challenges due to two fundamental properties of the data: high sparsity and the lack of paired ground truth observations (unpaired control and perturbed populations). To address this, we utilize OT as a preprocessing step to infer latent couplings, combined with MK-MMD or reconstruction losses.

To rigorously demonstrate the necessity of this approach compared to baselines (e.g., Random Pairing or MK-MMD-only objectives), we conducted an empirical ablation study using synthetic data. This controlled environment allows us to overcome the "missing pair" problem inherent in real biological data, where the lack of ground truth counterfactuals makes it impossible to definitively validate causal alignment. The code to reproduce these results is available in our source code.

We utilized synthetic data for this analysis for three critical reasons:

1. **Unobservable Counterfactuals:** In real scRNA-seq, measuring a cell destroys it, making it impossible to observe the same cell in both control and perturbed states ($x_i$ and $y_i$). Synthetic data allows us to fabricate ground truth pairs, providing access to latent ground truth to definitively quantify causal accuracy.

2. **Controlled Sparsity Sweep:** Real data has fixed, high sparsity. Synthetic data allows us to sweep sparsity from $0\%$ to $99\%$ to empirically observe the phase transition where reconstruction metrics degrade.

3. **Signal Isolation:** It allows us to isolate the mathematical properties of the loss functions from biological noise and batch effects.

We simulated a dataset of 1000 samples with 5000 features (genes).

- **Control Population ($X$):** Generated from $\mathcal{N}(0, 1)$ and masked to achieve target sparsity levels ranging from $0\%$ to $99\%$.

- **Perturbation:** A systematic shift was added to the first 50 features ($+2.0$) to simulate a biological effect.

- **Ground Truth ($Y$):** The perturbed state for each control cell, preserving the sparse structure.

We evaluated three hypothetical model outputs against the ground truth $Y$, representing the outcomes of different training strategies:

1. **Causal/Paired Model (Ideal):** The ground truth $Y$ with added Gaussian noise ($\sigma = 0.1$). *Representation:* This proxies the outcome of a model trained with OT-based pairing. By recovering the latent pairs ($x_i \rightarrow y_i$), the model learns the correct cell-specific trajectories.

2. **Population/Generative Model (Unpaired):** The ground truth population $Y$ with sample indices randomly permuted. This proxies the equilibrium state of a model trained with MK-MMD only. Such a model perfectly matches the target distribution $P(Y)$ (minimizing the MK-MMD loss) but fails to learn the causal mapping ($f(x_i) \neq y_i$), effectively becoming a generative model of the population rather than a causal predictor.

3. **Mode Collapse (Average):** The mean vector $\bar{Y}$ repeated for all samples. This represents the failure mode of Random Pairing (MSE). When training on random pairs, the model minimizes variance by predicting the population mean.

We evaluated three metrics in this extended results:

- **MSE:** Standard point-wise reconstruction loss.

- **MK-MMD:** Distributional distance using a multi-scale Gaussian kernel.

- **OT:** Exact EMD (Wasserstein-2).

### F.4.1 RESULTS

We evaluated the metrics across varying sparsity levels to identify the failure modes of standard losses.

Table S7: Metric Performance Across Sparsity Regimes. Lower is better. Bold indicates the metric's "preferred" model.

| Sparsity | Metric | Causal/Paired (OT+MK-MMD) | Population/Generative (MK-MMD Only) | Mode Collapse (MSE Failure) | Verdict |
|---|---|---|---|---|---|
| **0%** (Dense) | **MSE** **MK-MMD** | **0.0100** **−0.0061** | 1.9947 **−0.0062** | 1.0000 2.3127 | **Success.** **Ambiguity.** |
| **50%** (Medium) | **MSE** **MK-MMD** | **0.0100** −0.0059 | 0.9987 **−0.0062** | 0.4989 2.3119 | **Degradation.** **Failure.** |
| **90%** (High) | **MSE** **MK-MMD** | **0.0100** −0.0022 | 0.1997 **−0.0062** | 0.0997 2.3066 | **Weakness.** **Failure.** |
| **99%** (Extreme) | **MSE** **MK-MMD** **OT** | 0.0100 0.1648 49.95 | 0.0201 **−0.0061** **0.00** | **0.0101** 2.2505 50.26 | **Failure.** **Failure.** **Failure.** |

The most critical finding is observed in the "Population/Generative" column. Across all sparsity levels, MK-MMD assigns a perfect or near-perfect score ($\approx 0.0$) to the Population Model. This implies that a model trained with MK-MMD as the sole objective has no incentive to learn the correct causal mapping. It can achieve zero loss simply by generating a realistic population that is causally scrambled (i.e., Control Cell A is mapped to Perturbed State B). The result is that the model learns the *distribution* but loses the *cell identity*.

Even if we attempted to force pairing using standard reconstruction (MSE), the high sparsity (99%) causes MSE to favor the Mode Collapse (0.0101) over the correct structure (0.0100). This empirically demonstrates that training on random pairs forces the model to converge to the population mean, which on sparse data is indistinguishable from a valid prediction in terms of error magnitude.

The empirical results provide definitive proof that standard approaches are insufficient for sparse, unpaired causal inference:

1. **MK-MMD-Only Training** leads to Causal Scrambling (Permutation Invariance).
2. **Random Pairing / MSE Training** leads to Mode Collapse (Convergence to Mean).

This justifies the necessity of **OT preprocessing**. By solving for the optimal coupling matrix $\pi$ that minimizes transport cost, we explicitly enforce the Principle of Minimal Action, assuming that cells undergo the smallest necessary transcriptomic change. This effectively infers the latent pairing that MSE assumes exists, resolving the identifiability crisis that MK-MMD ignores and avoiding the collapse mode that Random Pairing induces.

### F.5 EXTENDED RESULTS: IMPACT OF OPTIMAL TRANSPORT ON DOWNSTREAM TASKS

While the primary motivation for integrating OT is to prevent mean collapse and preserve population structure during training, its ultimate value lies in improving performance on downstream biological tasks. We conducted an ablation study comparing the standard OT-based pairing against a Random Pairing baseline to isolate the impact of this preprocessing step on the model's ability to predict non-additive genetic interactions.

We evaluated both models on the "Genetic Interaction" benchmark (Precision @ 10). As shown in Table S8, the inclusion of OT leads to a marked improvement in identifying complex interaction types, specifically Redundancy, Epistasis, and Suppression.

Table S8: Ablation Study: Impact of Optimal Transport on Genetic Interaction Prediction (Precision @ 10). The model trained with OT preprocessing consistently outperforms the Random Pairing baseline in detecting complex interaction types (Redundancy, Epistasis, Suppression), demonstrating that OT helps preserve the fine-grained causal structure required for these tasks.

| Interaction Type | Random Pairing | Optimal Transport |
|---|---|---|
| Neomorphism | 0.4 | 0.4 |
| Redundancy | 0.7 | **0.8** |
| Epistasis | 0.8 | **0.9** |
| Synergy | **0.5** | 0.4 |
| Suppression | 0.2 | **0.3** |

The results indicate that while Random Pairing can achieve competitive performance on simpler metrics (like Neomorphism), it falls short in capturing the subtle dependencies required to identify Redundancy and Epistasis. The OT pairing, by matching cells based on their distributional similarity, likely preserves the underlying biological signal that defines these interactions, preventing them from being washed out by the noise of random assignment. This validates the hypothesis that OT is not just a theoretical necessity for causal identifiability but a practical enhancer of model utility for complex biological discovery.

### F.6 EXTENDED RESULTS: ABLATION STUDY ON $\beta$-VAE AND DAGMA REGULARIZATION

The primary goal of this ablation study is to rigorously assess the sensitivity of the RAPTORGraph model to its key regularization parameters, specifically the $\beta$ parameter of the $\beta$-VAE ($\beta$) and the weights of the DAGMA loss ($\mu$ and $\lambda_1$). Our core hypotheses for this study were: 1. The criticality of $\beta$ for preventing mode collapse, where we investigated if $\beta$ is the most crucial hyperparameter for maintaining a regularized and informative latent space, looking for evidence of KL divergence approaching zero and its detrimental effects on learning meaningful representations. 2. The hierarchy of hyperparameter importance, testing the assertion that $\beta$ is of primary importance, followed by the DAGMA weights, in terms of their impact on downstream performance.

The study was conducted using the RAPTORGraph model on the Norman et al. 2019 dataset. For the hyperparameter sweep, $\beta$ was swept across `[0.0, 0.1, 0.4, 0.8, 1.2, 1.6, 2.0, 2.5, 3.0]` to observe its effect on KL divergence and to test for mode collapse. Additionally, $\mu$ and $\lambda_1$ were varied around default values to analyze their influence on the causal graph structure, specifically $\mu \in$ `[0.1, 0.27, 0.5]` and $\lambda_1 \in$ `[0.02, 0.05, 0.1]`. The evaluation metrics included: a) KL Divergence as the primary metric to monitor for signs of mode collapse; b) Prediction Variance (Pred. Var.), representing the variance of the reconstructed gene expression for *control* cells, used to detect mean collapse; and c) MK-MMD to evaluate the quality of predictions for *intervened or perturbed* cells. Notably, MSE is not explicitly used for mode collapse detection in this context, as variance directly assesses the diversity of generated outputs, which MSE alone might obscure.

To better visualize the dominant effect of $\beta$ on the latent space, Table S9 summarizes all experiments. The goal is to demonstrate that $\beta$ is the primary factor driving the KL divergence towards zero regardless of the DAGMA hyperparameter settings, indicating posterior collapse.

The consolidated table makes the trend clear. For a given $\beta$ value (e.g., 0.4), the KL Divergence remains consistently low ($\approx 0.004$) across all variations of $\mu$ and $\lambda_1$. However, changing $\beta$ from 0.1 to 0.4, and then to 0.8 and higher, causes the KL Divergence to drop by orders of magnitude. This demonstrates that $\beta$ is the determining factor for the degree of regularization and subsequent posterior collapse. The DAGMA weights have a negligible effect on the KL Divergence itself, reinforcing the conclusion that an appropriate $\beta$ must be selected first before fine-tuning the other parameters.

Our argument is to choose $\beta = 0.4$ (approx. 0.447 in our final model) to balance the trade-off between distribution matching (MK-MMD) and preserving biological heterogeneity. The KL Divergence at this range ($\approx$ 1e-3) is chosen to be similar to baselines like dVAE and SENA, ensuring the model

Table S9: Impact of $\beta$-VAE and DAGMA Regularization on Model Performance. **Pred. Var.** denotes the variance of the predicted cell states (or variance of predicted control cells), where a lower value may indicate mean collapse. Lower MK-MMD indicates better distribution matching. The bold row indicates the selected best configuration.

| $\beta$ | $\mu$ | $\lambda_1$ | **KL Divergence** | **Pred. Var.** | **MK-MMD** |
|---|---|---|---|---|---|
| 0.0 | 0.27 | 0.052 | 0.0181 | 0.0181 | 1.929 |
| 0.1 | 0.1 | 0.02 | 0.059 | 0.0059 | 0.268 |
| 0.1 | 0.1 | 0.05 | 0.058 | 0.0067 | 0.257 |
| 0.1 | 0.1 | 0.1 | 0.054 | 0.0073 | 0.161 |
| 0.1 | 0.27 | 0.02 | 0.061 | 0.0051 | 0.401 |
| 0.1 | 0.27 | 0.05 | 0.060 | 0.0062 | 0.318 |
| 0.1 | 0.27 | 0.1 | 0.0615 | 0.0068 | 0.209 |
| 0.1 | 0.5 | 0.02 | 0.0609 | 0.0048 | 0.510 |
| 0.1 | 0.5 | 0.05 | 0.0604 | 0.0054 | 0.450 |
| 0.1 | 0.5 | 0.1 | 0.0572 | 0.0058 | 0.319 |
| 0.4 | 0.1 | 0.02 | 0.0040 | 0.0010 | 0.120 |
| 0.4 | 0.1 | 0.05 | 0.0039 | 0.0010 | 0.117 |
| 0.4 | 0.1 | 0.1 | 0.0040 | 0.0009 | 0.117 |
| 0.4 | 0.27 | 0.02 | 0.0040 | 0.0009 | 0.133 |
| **0.4** | **0.27** | **0.05** | **0.0039** | **0.0009** | **0.127** |
| 0.4 | 0.27 | 0.1 | 0.0040 | 0.0008 | 0.121 |
| 0.4 | 0.5 | 0.02 | 0.0040 | 0.0007 | 0.203 |
| 0.4 | 0.5 | 0.05 | 0.0040 | 0.0007 | 0.196 |
| 0.4 | 0.5 | 0.1 | 0.0040 | 0.0007 | 0.209 |
| 0.8 | 0.1 | 0.02 | 0.0009 | 0.0002 | 0.136 |
| 0.8 | 0.1 | 0.05 | 0.0009 | 0.0002 | 0.132 |
| 0.8 | 0.1 | 0.1 | 0.0009 | 0.0002 | 0.127 |
| 0.8 | 0.27 | 0.02 | 0.0009 | 0.0002 | 0.147 |
| 0.8 | 0.27 | 0.05 | 0.0009 | 0.0002 | 0.146 |
| 0.8 | 0.27 | 0.1 | 0.0009 | 0.0002 | 0.143 |
| 0.8 | 0.5 | 0.02 | 0.0008 | 0.0001 | 0.215 |
| 0.8 | 0.5 | 0.05 | 0.0009 | 0.0001 | 0.227 |
| 0.8 | 0.5 | 0.1 | 0.0009 | 0.0001 | 0.206 |
| 1.2 | 0.27 | 0.052 | 0.0003 | 0.00005 | 0.137 |
| 1.6 | 0.27 | 0.052 | 0.0002 | 0.00002 | 0.144 |
| 2.0 | 0.27 | 0.052 | 0.0001 | 0.00001 | 0.155 |
| 2.5 | 0.27 | 0.052 | 0.0001 | 0.000007 | 0.171 |
| 3.0 | 0.27 | 0.052 | 0.0000 | 0.000004 | 0.167 |

captures meaningful biological variability rather than regressing to the population mean. While $\beta = 0.1$ preserves more variance, it suffers from poor stability and high MK-MMD. Conversely, $\beta = 0.8$ achieves competitive MK-MMD but suffers from an order-of-magnitude drop in prediction variance (1e-4 vs 1e-3), indicating severe over-smoothing. We select $\beta = 0.4$ as the optimal operating point, minimizing MK-MMD while retaining significantly higher variance than the fully collapsed regimes. The bolded row in Table S9 represents this chosen configuration.

# G  LARGE LANGUAGE MODEL (LLM) USAGE DISCLOSURE

## G.1  DECLARATION OF AI-ASSISTED TECHNOLOGIES USAGE

In compliance with ICLR 2026 policies on Large Language Model usage, we disclose the following usage of AI-assisted tools in the preparation of this manuscript.

### G.1.1  LANGUAGE EDITING AND POLISH

**Tool:** DeepL Write and Google Gemini 2.5 Pro.

**Purpose:** We utilized these tools for grammar correction, style improvement, and enhancing the overall polish of the manuscript text.

**Usage:** The process began with the authors drafting the initial text, establishing the core scientific narrative and technical details. Following the initial draft, selected sections were then refined using the LLMs to improve clarity, flow, and grammatical correctness. All technical content, research findings, and conclusions remain entirely authored by the human researchers.

### G.1.2  CODE DEVELOPMENT AND IMPLEMENTATION

**Tool:** Google Gemini 2.5 Pro.

**Purpose:** The LLM was used to assist with code development, debugging, and the implementation of research algorithms.

**Usage:** The authors first designed the overall software architecture and wrote the initial prototype code. The LLM was then used interactively to help generate boilerplate code snippets and assist with debugging specific functions. This assistance was particularly helpful for making the code more robust and efficient. All code was thoroughly reviewed, tested, and validated by the authors to ensure correctness and alignment with our research objectives.

## G.2  AUTHOR RESPONSIBILITY STATEMENT

In accordance with ICLR 2026 Policy 2, the authors take full responsibility for all content in this submission. All AI-generated content has been carefully reviewed, verified, and validated by the human authors. The authors certify that:

- No factual claims were accepted from LLMs without independent verification.
- All experimental results and data analysis were conducted by human researchers.
- The core research contributions and insights are entirely the work of the human authors.
- Proper attribution has been provided for all external sources and previous work.

The use of these AI tools was solely to enhance the clarity, efficiency, and quality of our research presentation while maintaining the integrity and originality of our scientific contributions.

