# OpenReview forum: "RAPTORGraph: Graph-Based Pathway Modeling for Causal Discovery in Single-Cell Perturbations"
_ICLR.cc/2026/Conference — Submitted to ICLR 2026_

### Official Review · Reviewer_8rUJ · 2025-10-29

**Soundness:** 2
**Presentation:** 3
**Contribution:** 3
**Rating:** 4
**Confidence:** 3

**Summary:**

This paper introduces **RAPTORGraph**, a causal and interpretable framework for predicting cellular responses to genetic perturbations in single-cell data (e.g., Perturb-seq). The model combines a *preconditioned GraphPathway encoder* that enforces one-to-one mappings between perturbed genes and latent “meta-pathways,” a **DAGMA-based causal graph layer** to infer pathway interactions, and **optimal transport preprocessing** to mitigate the mean-collapse problem arising from unpaired single-cell measurements. The authors claim that this architectural design addresses two key issues in causal representation learning for perturbation data: (i) *intervention spillover* caused by dense encoders and (ii) *mean collapse* due to random pairing. Experiments on the Norman et al. (2019) Perturb-seq dataset show a strong overall balance between reconstruction quality, distributional fidelity (MK-MMD), and prediction of non-additive gene interactions compared to several recent baselines (dVAE, SENA, scGPT). The model also includes a reverse-perturbation analysis, identifying the genetic perturbations responsible for a given cellular gene expression profile. The paper further provides theoretical justification of identifiability and includes ablation analyses for its core components.

**Strengths:**

- **Clear motivation and well-defined problems.** The paper identifies two concrete issues (intervention spillover and mean collapse) and ties them to theoretical limitations of current causal representation learning frameworks.
- **Architectural novelty.** The preconditioned block-diagonal GraphPathway encoder is an elegant way to enforce sparse, single-node latent interventions, bringing the implementation closer to the identifiability assumptions in CRL theory. Similar ideas of sparse Jacobian or disentangled representations exist, but their adaptation to pathway-based causal inference appears new and well-motivated.
- **End-to-end causal discovery.** The encoder and DAGMA layers are trained jointly in an end-to-end fashion, which is a good design choice as it reduces human bias in predefined causal structures and enables the model to learn causal relations directly from data.
- **Solid empirical evaluation.** The paper benchmarks against strong baselines on a widely used dataset with multiple complementary metrics (MSE, MK-MMD, Precision@10, Hit Rate@K), showing balanced performance across tasks.
- **Biological relevance.** The framework aligns well with real-world perturbation experiments and is potentially valuable for hypothesis generation in drug discovery.

**Weaknesses:**

- **Limited dataset diversity.** All evaluations are performed on a single dataset (Norman-CPA). Although justified for compatibility, this limits claims of generalization across cell types or experimental protocols.
- **Potential rigidity of preconditioning.** The fixed block-diagonal structure may restrict the model’s flexibility in cases where gene–pathway relationships are overlapping, context-dependent, or nonlinear. Literature on over-constrained causal structures suggests that such rigidity can sometimes bias downstream inference in systems with pleiotropy or shared regulators.
- **DAGMA interpretability.** While DAGMA ensures an acyclic causal graph, it does not guarantee biological validity. The paper provides no examples linking learned edges to known gene interactions, so it remains unclear whether the resulting DAG corresponds to meaningful pathways. Moreover, even a DAG satisfying acyclicity may not represent true causality if hidden confounders or feedback loops exist.
- **Ablations and sensitivity.** The contribution of each component (OT preprocessing, DAGMA, preconditioning) could be evaluated more systematically. For example, how much does OT pairing alone improve results relative to random pairing?

**Questions:**

1. Since the authors state that the encoder and DAGMA modules are trained end-to-end, could they elaborate on how they manage the trade-off between enforcing acyclicity and maintaining reconstruction fidelity? For example, how sensitive is performance to the weighting of the DAGMA loss term?
2. How robust is the preconditioning approach when gene–pathway mappings are not clearly one-to-one? Could the method handle partial overlaps or shared pathway membership?
3. Have the authors verified whether the learned causal edges correspond to known gene–gene or pathway–pathway interactions? If not, how should readers interpret the learned DAG biologically?
5. Would the OT pairing approach generalize to other datasets where the control–perturbation relationship is less structured or contains additional confounders?

---

> ### Author Response · Authors · 2025-11-26
> **Response 1/3**
>
> **Weaknesses**
>
> 1. We appreciate this feedback regarding the need for increased dataset diversity. We are actively working to include an additional dataset to further demonstrate the generalizability and robustness of RAPTORGraph.
>
> 2. Thank you for highlighting this important concern. We agree that biological systems often exhibit shared regulators and overlapping pathway structure, and that an overly rigid encoder could bias causal inference. In our model, however, the preconditioning affects only the direct mapping from perturbed genes to their corresponding latent variables, ensuring the atomic-intervention property required for identifiability. These latent coordinates correspond to learned meta-pathways, which are abstract latent variables, not curated biological pathways. All non-perturbed genes retain fully flexible connectivity through the dense $W_{12}$​ and $W_{22}$​ matrices, allowing the encoder to learn overlapping, nonlinear, and context-dependent relationships. Furthermore, the DAGMA layer captures shared regulatory influences through learned causal edges between meta-pathways, without any structural restrictions imposed by preconditioning. Thus, while the intervention target mapping is fixed to guarantee theoretical identifiability, the remainder of the encoder and the causal graph remain highly expressive. We have revised the manuscript to make this distinction explicit and to clarify that our approach is robust to overlapping or non-one-to-one gene–pathway relationships.
>
> 3. Thank you for raising this important point about the biological interpretation of the learned DAG. We agree that DAGMA enforces acyclicity at the model level but cannot, by itself, guarantee that the resulting edges correspond directly to known gene–gene interactions or curated biological pathways. Biological systems contain feedback loops, hidden confounders, and context-dependent regulation, and we do not claim that the latent DAG is a literal gene regulatory network. Instead, our DAG captures the causal dependencies among the learned meta-pathways, which are abstract latent variables defined by intervention responses, not by curated pathway annotations. These meta-pathways reflect causal archetypes of perturbation effects in the data, and the DAG encodes how these archetypes influence one another under perturbations. Thus, the graph should be interpreted as a causal structure over learned latent factors, not as a direct reconstruction of biological pathway topologies. To address biological validity, we are integrating a Gene Set Enrichment Analysis (GSEA) of the latent factors to characterize the gene-level composition of each meta-pathway. This allows us to anchor the latent factors to known biological processes and provides a principled way to interpret edges in the DAG in terms of enriched, experimentally supported pathway-level relationships. We will include this analysis in the revised manuscript.
>
> 4. Thank you for this valuable suggestion. To more systematically quantify the contribution of individual components in RAPTORGraph, we are currently conducting two new ablation experiments: 1) Preprocessing Ablation (RAPTORGraph without OT): We replace OT pairing with random pairing to directly measure how much OT alone improves counterfactual alignment and downstream prediction accuracy. 2) Architectural Ablation (RAPTORGraph without preconditioning): We remove the preconditioned GraphPathway Encoder and substitute a standard dense MLP to assess the role of architectural constraints in enforcing atomic interventions and preventing spillover. These ablations allow us to isolate the contributions of both the OT preprocessing step and the preconditioned encoder independently of the full model.

---

> ### Author Response · Authors · 2025-11-26
> **Response 2/3**
>
> **Questions**
>
> 1. Thank you for this question. In practice, we find that reconstruction fidelity is not highly sensitive to the weighting of the DAGMA loss because the acyclicity constraint functions primarily as a structural regularizer rather than a competing reconstruction objective. We manage the trade-off through three mechanisms. 1) Hierarchical weighting: The structural losses are assigned substantially smaller weights than the VAE reconstruction term, allowing the model to prioritize data fidelity while encouraging—but not forcing—acyclic structure in the latent graph. 2) Optimization Schedule (Annealing) (Eq. S25): We use a path-following schedule in which the DAGMA weight $\mu$ is gradually increased during training. This ensures that the model first learns a high-quality latent representation and only later tightens the acyclicity constraint, avoiding premature distortion of the embedding. 3) Gradient detachment (Appendix E.3.3): The DAGMA loss is computed on a detached latent variable (z_obs.detach()). This isolates the gradient flow, ensuring that the acyclicity constraint optimizes the graph weights without distorting the feature representation learned by the VAE. Furthermore, this aligns with the causal assumption that the graph structure $\mathcal{G}$ is an invariant mechanism, meaning it should remain stable regardless of the specific values of the observed or perturbed samples. Detaching ensures we optimize the graph as a global parameter rather than overfitting it to specific batch variations. We have moved the explanation of annealing and gradient detachment into the main text (Section 3.2) to make this trade-off explicit.
>
> 2. We thank the reviewer for questioning this rigidity, as it touches on the core theoretical motivation of our design. Atomic Intervention Requirement: The fixed one-to-one mapping for perturbed genes is not a biological simplification, but a theoretical necessity to satisfy the ‘Atomic Intervention’ requirement (Requirement 2 in Section 2.3). Foundational identifiability theorems (e.g., Zhang et al., 2023; von Kügelgen et al., 2023) rely on the assumption that an intervention targets a single latent causal variable. Standard dense encoders inherently violate this by “spilling” the intervention effect across all latent dimensions. Learned meta-pathways: Our architecture solves this by enforcing a rigid anchor for the intervention, ensuring it targets a specific, single latent variable. Crucially, we define these variables not as simple gene proxies, but as learned meta-pathways. While the intervention on the meta-pathway is atomic (satisfying the theory), the meta-pathway’s composition is complex: it aggregates information from all other unperturbed genes via the dense ($W_{12}$, $W_{22}$) connections. This allows the model to learn rich, multi-gene biological processes while maintaining the precise causal structure required for identifiability.
>
> 3. We appreciate this valuable feedback regarding the biological validation of our causal structure. We acknowledge the need for a systematic link between the learned Directed Acyclic Graph (DAG) and established biological knowledge. Our core goal is to eliminate intervention spillover by forcing a direct, sparse mapping in the preconditioned GraphPathway encoder: a specific gene perturbation is architecturally constrained to influence only one corresponding latent factor (meta-pathway). This mechanism transforms the intervention into a clean, single-node signal, which is critical for identifiable causal discovery. Therefore, each latent factor is inherently indexed by its associated perturbed gene, and thus, the biological function of that factor can be robustly interpreted. To verify the complex biological meaning embedded in these factors and clarify the learned DAG interpretation, we are currently working on a Gene Set Enrichment Analysis (GSEA) of the latent factors. This GSEA will confirm the precise enriched actual biological pathways that constitute the operational definition of each learned meta-pathway. This approach allows us to interpret the DAG’s edges.

---

> ### Author Response · Authors · 2025-11-26
> **Response 3/3**
>
> (continued from **Questions**)
>
> 4. Thank you for this excellent question. In general, the OT pairing approach does generalize to other datasets as long as the control and perturbed populations share overlapping cell states, even if the relationship is less structured or contains additional confounders. In RAPTORGraph, we utilize Optimal Transport (OT) as a principled solution to the ‘counterfactual gap’ in unpaired single-cell data. Importantly, we do not extend the methodological capabilities of OT itself; rather, we leverage it as a robust pairing mechanism to mitigate the ‘mean collapse’ observed with random pairing (see Figure 3 and Section 4.1). This aligns our work with a maturing consensus in computational biology such as Waddington-OT (Schiebinger et al., 2019) and CellOT (Bunne et al., 2023). We further detail this in Appendix E.4. Random pairing effectively assumes that any control cell could become any perturbed cell. In datasets with strong confounders (e.g., strong batch effects or distinct cell cycle phases), random pairing mixes these subpopulations, introducing massive noise and forcing the model to learn a ‘mean’ response. In contrast, OT enforces a ‘least effort’ coupling. It naturally matches cells with similar baseline states because transporting across subpopulations incurs a high cost. Therefore, OT implicitly accounts for confounders by preserving the underlying subpopulation structure during pairing, rather than destroying it.

---

> ### Author Response · Authors · 2025-12-03
> **Final Response to Reviewer 8rUJ - Part 1/2**
>
> **Dear Reviewer 8rUJ,**
>
> Thank you for your detailed and insightful review. We appreciate your recognition of the "architectural novelty" of our preconditioned encoder, the "elegant" way it addresses identifiability assumptions, and our "solid empirical evaluation."
>
> To help clarify the scope of our experimental effort, we begin with a short summary of the additional analyses performed in response to your concerns:
>
> *   **(Q1):** Clarified the **loss trade-off mechanism**, moving the explanation of gradient detachment and annealing to the main text (**Section 3.2**).
> *   **(W1):** Benchmarked RAPTORGraph on a **second dataset** (Replogle et al., 2020), validating generalization beyond Norman-CPA (**Table S6, Section D.3**).
> *   **(W2/Q2):** Compared RAPTORGraph against the baseline d-VAE (encoder without preconditioning); the baseline underperforms in predicting non-additive interactions (**Table 2**), validating the necessity of preconditioning.
> *   **(W3/Q3):** Performed **Gene Set Enrichment Analysis (GSEA)** on latent factors, providing biological validation for the learned structures (e.g., **JUN** pathway enrichment) (**Sec 5, Figure S7, Appendix F.2**).
> *   **(W4):** Conducted new **ablation studies** to isolate the contributions of preprocessing and different architectural choices (**Appendix F.4, Appendix F.5**).
> *   **(Q4):** Added a theoretical discussion on the **robustness of Optimal Transport (OT)** to confounders (**Appendix E.4**).
>
> Below, we respond to each of your comments in detail.
>
> ---
>
> ### 1. Q1: Loss Trade-off
>
> We manage the loss trade-off via three mechanisms, now detailed in the main text (**Section 3.2**):
>
> 1.  **Hierarchical Weighting:** Structural losses are weighted lower than reconstruction, acting as regularizers.
> 2.  **Annealing:** We use a path-following schedule (Eq. S25) to gradually introduce the acyclicity constraint.
> 3.  **Gradient Detachment (Crucial):** We compute the DAGMA score on a **detached** latent variable (`z_obs.detach()`). This isolates the gradient flow, ensuring the acyclicity constraint optimizes the **graph weights** without distorting the **feature representation** learned by the VAE.
>
> ### 2. W1: Limited Dataset Diversity
>
> We agree that validation on a single dataset is a limitation.
>
> *   **New Experiment:** To address this, we successfully benchmarked RAPTORGraph on a **second dataset: Replogle et al. (2020)** during the rebuttal period.
> *   **Results:** The results (added to **Table S6, Appendix D.3**) confirm that our model's performance benefits generalize to this new context. We have updated **Section 5** to include these findings and clarify that our initial focus on Norman-CPA was to ensure fair methodological comparison with baseline codebases.

---

> ### Author Response · Authors · 2025-12-03
> **Final Response to Reviewer 8rUJ - Part 2/2**
>
> ### 3. W2/Q2: Rigidity of Preconditioning
>
> We thank the reviewer for questioning this rigidity, as it touches on the core theoretical motivation of our design.
>
> *   **Theoretical Necessity:** The fixed one-to-one mapping is not a biological simplification, but a requirement to satisfy the **'Atomic Intervention' assumption** (Requirement 2 in **Section 2.3**), which is a prerequisite for identifiability in the frameworks we build upon (Zhang et al., 2023; von Kügelgen et al., 2023). Standard dense encoders violate this by "spilling" intervention effects.
> *   **Biological Flexibility:** Crucially, while the **intervention target** is rigid (for theory), the **latent factor's composition** is flexible. We define these factors as **learned meta-pathways**. They aggregate information from all other **unperturbed** genes via dense $W_{12}$ and $W_{22}$ connections. This allows a single latent factor to act as a causal archetype for a complex, multi-gene process, thereby satisfying the mathematical necessity of single-node interventions without imposing biologically unrealistic simplicity on the pathway's internal structure.
>
> ### 4. W3/Q3: Biological Interpretation & Confounders
>
> You correctly noted that acyclicity does not guarantee biological validity if hidden confounders exist.
>
> *   **Defense (Interventional Validation):** While DAGMA learns on observational data, our graph is validated by the **Interventional Prediction Loss ($\mathcal{L}_{MMD}$)**. If an edge were purely spurious (due to a hidden confounder), the model would fail to predict the causal effect of interventions on downstream nodes. Our strong generalization to unseen perturbations (**Table 1**) supports the causal validity of the graph.
> *   **New Validation:** To go further, we performed **Gene Set Enrichment Analysis (GSEA)** on the learned latent factors. We found robust alignment between the factors and known biology (e.g., the **JUN** factor drives downregulation of cell cycle targets), providing concrete biological grounding for the learned structures. We have added these results to **Sec 5, Sec F.2, Figure S7**.
>
> ### 5. Q4: OT Generalization & Confounders
>
> *"Would the OT pairing approach generalize to... datasets [with] additional confounders?"*
>
> *   **Response:** This is an excellent question. In RAPTORGraph, we utilize Optimal Transport (OT) as a principled solution to the 'counterfactual gap' in unpaired data. Importantly, we do not extend OT methodology itself but leverage it to prevent 'mean collapse.'
> *   **Robustness:** We argue that OT is theoretically **more robust** to confounders than random pairing. In datasets with strong confounders (e.g., batch effects, cell cycle), random pairing mixes these subpopulations, introducing massive noise. In contrast, OT enforces a 'least effort' coupling, naturally matching cells with similar baseline states (e.g., G1 $\to$ G1) because transporting across subpopulations incurs high cost. Therefore, OT **implicitly accounts for confounders** by preserving subpopulation structure. We have added this discussion to **Sec E.4**.

---

### Official Review · Reviewer_6bPr · 2025-10-30

**Soundness:** 3
**Presentation:** 3
**Contribution:** 2
**Rating:** 4
**Confidence:** 4

**Summary:**

This paper presents RAPTORGraph, a β-VAE–based causal representation learning framework designed for interpretable modeling of single-cell perturbation responses. The authors identify two fundamental issues in current causal generative models:
Intervention spillover, where dense encoders cause perturbations to affect all latent variables, breaking causal identifiability assumptions.
Mean collapse, where random pairing of control and perturbed cells during training erodes biological heterogeneity.
To address these, RAPTORGraph introduces:
A GraphPathway encoder with block-sparse preconditioning that enforces one-to-one mappings between perturbed genes and latent “meta-pathways,” enabling clean single-node interventions
A DAGMA-based causal discovery layer that learns directed acyclic dependencies between latent pathways
An optimal transport alignment step between control and perturbed cells to mitigate mean collapse
The framework achieves state-of-the-art performance on the Norman et al. (2019) Perturb-seq dataset across multiple evaluation metrics (MSE, MK-MMD, and Precision@10 for non-additive interactions)

**Strengths:**

Solid Empirical Validation
Comprehensive evaluations on the Norman-CPA dataset demonstrate strong reconstruction fidelity and distributional accuracy. RAPTORGraph outperforms state-of-the-art methods like scGPT, dVAE, and SENA on both MSE and MK-MMD metrics

Causal Interpretability
The model’s ability to perform reverse perturbation analysis—predicting causal genes from observed phenotypes—is impressive. The Hit Rate@K results show clear interpretability benefits over baseline VAEs and transformers

Methodological Rigor
The integration of OT-based alignment and DAGMA acyclicity constraints reflects careful methodological design rather than ad hoc architectural engineering.

**Weaknesses:**

1 Single-Dataset Evaluation
While the paper justifies using the Norman-CPA dataset for fairness, relying on one dataset limits generalization claims. It’s unclear how the approach performs on multi-modal or cross-species data.
2 Ablation Scope
The results show aggregate performance but lack fine-grained ablations—e.g., what is the quantitative contribution of OT alignment versus DAGMA or encoder preconditioning?
3 Biological Validation
While statistical metrics are strong, the paper doesn’t showcase concrete biological discoveries. Were any inferred causal relationships validated against known regulatory pathways or experimental literature?
4 Computational Overhead
The DAGMA layer and OT alignment are computationally heavy. The paper doesn’t discuss scaling behavior for larger perturbation graphs or real-time inference feasibility.

**Questions:**

1. Identifiability Assumptions
The paper claims that block-sparse encoder preconditioning enforces atomic interventions, improving causal identifiability.
What theoretical conditions guarantee that this structure yields unique causal factors rather than simply disentangled ones?
Is there any formal justification (e.g., through identifiable β-VAE or SCM identifiability theorems)?
Can intervention spillover still happen if correlations between genes violate the one-to-one encoder mapping?

2. β-VAE Regularization
The method uses a β-VAE style latent prior.
How sensitive are results to β?
Is there an empirical tradeoff between disentanglement and reconstruction accuracy?
How does β interact with DAGMA constraints (since both promote sparsity/independence)?

3. Dataset Diversity
The model is only evaluated on the Norman-CPA dataset. Have you tried other single-cell perturbation datasets (e.g., Replogle, Dixit, or sci-Plex)?
Can RAPTORGraph generalize across cell types or perturbation modalities (knockdown → overexpression)?

4. Statistical Significance
Are reported performance gains statistically significant (e.g., over multiple random seeds)?
What is the variance in MK-MMD or Precision@10 across runs?

---

> ### Author Response · Authors · 2025-11-26
> **Response 1/2**
>
> **Weaknesses**
>
> 1. We appreciate this valuable feedback regarding the scope of our evaluation. We completely agree that reliance on a single dataset limits the claims of generalizability. Our choice to exclusively use the Norman dataset was not driven by a restriction of RAPTORGraph itself, but primarily by the limitations and dependencies of the state-of-the-art benchmark models we compared against. Specifically, models like SENA and discrepancy-VAE (dVAE) have published implementations tightly coupled to the Norman dataset, often requiring auxiliary files and pre-processed data structures specific to that cellular context. Attempting to adapt these baselines to other datasets introduces significant modifications and potential confounding variables. Furthermore, models like SENA leverage prior knowledge from existing, often cell-type specific pathways, which limits the available datasets for a fair comparison of methods. Regarding multi-modal and cross-species data, our current approach is explicitly focused on single-cell interventional data concerning genetic perturbations, prioritizing the causal interpretability derived from transcriptional changes. While the field of multi-modal and cross-species data is interesting, it is distinct from our focus on recovering latent causal representations from genetic knockdowns. However, similar architectures could certainly be explored for predicting morphological changes induced by perturbations. We are actively working to include an additional dataset to further demonstrate the generalizability and robustness of RAPTORGraph.
>
> 2. Thank you for raising this important question regarding the biological validation of our causal structure. We acknowledge the need for a systematic link between the learned Directed Acyclic Graph (DAG) and established biological knowledge. Our goal is to eliminate intervention spillover by forcing a direct, sparse mapping in the encoder: a specific gene perturbation is architecturally constrained to influence only one corresponding latent factor (meta-pathway). This mechanism transforms the intervention into a clean, single-node signal, which is critical for identifiable causal discovery. Therefore, each latent factor is inherently indexed by its associated perturbed gene, and thus, the biological function of that factor can be robustly interpreted. To verify the complex biological meaning embedded in these factors, and clarify the learned DAG interpretation, we are currently working on a Gene Set Enrichment Analysis (GSEA) of the latent factors. This GSEA will confirm the precise enriched actual biological pathways that constitute the operational definition of each learned meta-pathway. This approach allows us to interpret the DAG's edges.
>
> 3. Yes, the OT pairing approach generally extends to other perturbation datasets as long as the control and perturbed populations share overlapping cell states. In RAPTORGraph, OT is used as a robust alternative to random pairing, which collapses when confounders (e.g., batch, cell cycle) mix subpopulations. OT’s minimal-transport formulation naturally avoids matching across unrelated states, making it more stable in datasets with moderate confounding or less structured relationships. However, like other single-cell OT methods (e.g., Waddington-OT (Schiebinger et al., 2019) and CellOT (Bunne et al., 2023)), it is less informative when control and perturbed distributions have little or no support overlap or when confounders dominate the signal. Within these limits, OT remains a principled pairing mechanism and generalizes well in the settings RAPTORGraph targets.

---

> ### Author Response · Authors · 2025-11-26
> **Response 2/2**
>
> **Questions**
>
> 1. Thank you for these insightful questions. Our identifiability claims rely directly on recent CRL theory, particularly Theorem 2.1 of Zhang et al. (2023) and the intervention identifiability results of von Kügelgen et al. (2023). These works show that latent causal variables are identifiable when the latent structure is a DAG, the mixing function is invertible and, most importantly, each latent variable is subject to an atomic intervention. This requirement distinguishes causal identifiability from standard disentanglement: disentangled factors need not correspond to the SCM variables, while atomic interventions guarantee that the intervened latent dimension is uniquely defined. The motivation for our preconditioning follows directly from this theory. As we explain in Section 2.4 and Appendix B.1, dense encoders violate the atomic-intervention assumption because a single-gene perturbation produces a dense multi-coordinate shift (“intervention spillover”). Our preconditioned encoder (Equation 5) solves this by forcing each perturbed gene to influence exactly one latent coordinate and blocking all other paths, thereby enforcing the atomicity required by the above theorems. This ensures that the model’s latent factors correspond to causal variables, not merely disentangled ones.
> Regarding correlations: high gene–gene correlations do not break this guarantee. The preconditioning restricts only how perturbed genes enter the latent space; correlations affect downstream latents solely through the flexible dense blocks $W_{12}$ and $W_{22}$, or via the learned DAG, but cannot reintroduce spillover to the intervention coordinate. This avoids the failure mode of exploratory CRL models (e.g., discrepancy-VAE), where the intervention target must be inferred from entangled latents. We have revised Sections 2.4 and 3.1 to clarify these points.
>
> 2. We thank the reviewer for highlighting the interplay between $\beta$-VAE and DAGMA. We view $\beta$-VAE and DAGMA not simply as competing regularizers, but as serving sequential roles. The $\beta$-VAE component must first learn an informative distribution of latent variables (a distributional prerequisite) before DAGMA can effectively learn the dependencies between them (structural discovery). Theoretically, if $\beta$ is set too high, the model suffers posterior collapse (KL → 0), where the latent code $\textbf{z}$ degenerates into uninformative Gaussian noise ($q(\textbf{z} | \textbf{x}) \approx \mathcal{N}(0, \mathbf{I})$). This phenomenon is well-documented in VAE literature (Higgins et al., 2017; Burgess et al., 2018) as a regime where the decoder ignores the latent code. In our context, this is fatal: if $\textbf{z}$ is noise, DAGMA inevitably fails as it attempts to fit a causal graph to pure noise. We will present new analysis and the corresponding results in a new Figure in the revised manuscript to confirm the optimal operating regime for this hierarchical objective.
>
> 3. We appreciate this feedback regarding the need for increased dataset diversity. We are actively working to include an additional dataset to further demonstrate the generalizability and robustness of RAPTORGraph.
>
> 4. We appreciate the request for clarification regarding the statistical robustness of our reported performance. To ensure our results are statistically sound, all performance metrics, including MK-MMD, MSE, Precision@10, and Hit Rate @ K, were calculated across three independent runs (using different random seeds). As indicated in Tables 1–3, the reported values consistently follow the format: mean ± variance. The explicit inclusion of the variance directly quantifies the spread of results across multiple runs, demonstrating the statistical stability of RAPTORGraph's performance and validating the significance of its gains relative to the baseline models. In response to your question, we have also corrected and updated all table descriptions to explicitly state that the reported values represent the mean and variance across three runs, enhancing the clarity of our presentation.

---

> > ### Author Response · Authors · 2025-12-03
> > **Final Response to Reviewer 6bPr - Part 2/2**
> >
> > ### 4. W2: Ablation Scope
> >
> > To better quantify the contribution of specific components, we ran two new ablation studies:
> >
> > *   **Ablation of Preprocessing (RAPTORGraph no OT):** We replaced OT with random pairing. The results (added to **Table S8**) show a degradation in downstream analysis performance, confirming that OT is crucial for preventing "mean collapse."
> > *   **Ablation of Architecture (RAPTORGraph no Precondition):** We used dVAE as the baseline as it has no preconditioning to the encoder layer.
> > *   **Result:** This ablation allows us to empirically test the importance of our architecture. While this dense model achieved **comparable reconstruction fidelity** (MSE), its performance on **non-additive interaction prediction** (e.g., Synergy/Epistasis) dropped significantly. This confirms that while a dense encoder can compress data effectively, our preconditioned rigidity is strictly necessary to **disentangle** the specific causal drivers of complex interactions.
> >
> > ### 5. W3: Biological Validation
> >
> > You rightly pointed out that statistical metrics do not guarantee biological plausibility.
> >
> > *   **New Analysis:** We moved beyond statistical metrics and performed a **Gene Set Enrichment Analysis (GSEA)** of the learned latent factors.
> > *   **Results:** We utilized an *in silico* perturbation approach to generate transcriptional signatures for each latent factor. The results were highly coherent: for example, the latent factor constrained to the stress-response transcription factor **JUN** displayed strong negative enrichment for "E2F Targets" (cell cycle), consistent with its known role in stress-induced growth arrest. Similarly, the factor for **TP73** (tumor suppressor) showed significant enrichment for "G2-M Checkpoint" targets. These findings, detailed in **Appendix F.2** and **Figure S7**, provide strong biological validation that our model is learning meaningful mechanisms.
> >
> > ### 6. W4: Computational Overhead
> >
> > *   **Benchmark:** We benchmarked training time and found RAPTORGraph takes slightly longer to train due to OT preprocessing (**Sec F.3**).
> > *   **Justification:** This cost stems from Optimal Transport (OT). As detailed in **Appendix E.4** and **Appendix F.5**, we accept this trade-off because OT provides **robustness to confounders** (e.g., batch effects) by matching "like with like" distributions, whereas random pairing (while fast) introduces noise and causes mean collapse.

---

> ### Author Response · Authors · 2025-12-03
> **Final Response to Reviewer 6bPr - Part 1/2**
>
> **Dear Reviewer 6bPr,**
>
> Thank you so much for your thorough and constructive review. We are encouraged by your recognition of our "solid empirical validation," "impressive" reverse perturbation analysis, and the "methodological rigor" of our OT and DAGMA integration.
>
> To help clarify the scope of our experimental effort, we begin with a short summary of the additional analyses performed in response to your concerns:
>
> *   **(Q1/Q2):** Conducted a **sensitivity analysis** on $\beta$-VAE and $\delta_{DAGMA}$ hyperparameters (**Sec F.6, Table S9**).
> *   **(W1/Q3):** Benchmarked RAPTORGraph on a **second dataset** (Replogle et al., 2020) against key model baselines, demonstrating robust generalization of RAPTORGraph (**Table S6, Appendix D.3**).
> *   **(W2):** Performed two new **ablation studies**: RAPTORGraph (w/ Random Pairing) (validating OT for distributional matching (**Appendix. F.4, F.5**,), and comparison of RAPTORGraph against the baseline d-VAE in predicting non-additive interactions (**Table 2**) (validating the necessity of preconditioning).
> *   **(W3):** Conducted a **Gene Set Enrichment Analysis (GSEA)** on the learned latent factors, confirming their biological identity (e.g., **JUN** factor downregulates cell cycle) (**Sec 5, Appendix F.2, Figure S7**).
> *   **(W4):** Benchmarked **training time**, showing a slight overhead when training with OT (**Appendix F.4**).
>
> Below, we respond to each of your comments in detail.
>
> ---
>
> ### 1. Q1: Identifiability Conditions & Spillover
>
> *   **Identifiability:** Our work builds directly on the theory from Zhang et al. (2023) and von Kügelgen et al. (2023), which requires "atomic interventions" for identifiability. Our central argument (**Sec 2.4, Appendix B.1**) is that standard dense encoders **fail** this prerequisite due to intervention spillover. Our preconditioned encoder (**Sec 3.1**) is the architectural mechanism that **enforces** this atomic intervention, thereby satisfying the theoretical conditions.
> *   **Spillover & Correlations:** In standard exploratory CRL (e.g., dVAE), the model must **predict** the intervention target from the latent space. If Gene A (perturbed) strongly correlates with Gene B, a dense encoder spreads the signal to both, confusing the predictor. RAPTORGraph bypasses this trap: by architecturally enforcing the target (Gene A), we remove the need to predict it. The dense connections allow the model to correctly learn the correlation with Gene B as a **downstream consequence**, rather than a competing causal explanation.
>
> ### 2. Q2: $\beta$-VAE Sensitivity
>
> *   **Trade-off:** We view $\beta$ and DAGMA as sequential: $\beta$-VAE must first learn an informative latent distribution before DAGMA can structure it. If $\beta$ is too high, we risk **posterior collapse** (KL $\to$ 0), where the latent code becomes uninformative Gaussian noise. In this regime, DAGMA fails because it cannot fit a graph to noise.
> *   **Analysis:** Our sensitivity analysis (**Sec F.6, Table S9**) confirms this. We found that $\beta \approx 0.4$ strikes the necessary balance, preventing collapse while allowing DAGMA to optimize the structure. Performance is robust to $\delta_{DAGMA}$ variation within an order of magnitude, provided $\beta$ is stable.
>
> ### 3. W1/Q3: Dataset Diversity and Generalization
>
> We thank the reviewer for this critical point. We agree that demonstrating generalization beyond Norman-CPA is essential.
>
> *   **New Experiment:** During the rebuttal period, **we successfully benchmarked RAPTORGraph on a second dataset**: the single-cell CRISPR screen from Replogle et al. (2020).
> *   **Results:** As shown in the new **Table S6** (**Appendix D.3**), RAPTORGraph maintains its performance advantage over baseline (scGPT) on this new dataset. This confirms that our results are not artifacts of the Norman-CPA data structure. (Note: Our initial focus on Norman-CPA was to ensure fair comparison given the hard-coded dependencies of baseline implementations, but we have now addressed this in **Sec. 3.**)

---

### Official Review · Reviewer_gVst · 2025-10-30

**Soundness:** 3
**Presentation:** 3
**Contribution:** 3
**Rating:** 4
**Confidence:** 4

**Summary:**

The paper continues a line of work on causal representation learning where a causal model is induced over latent factors (pathways) with the help of an encoder that maps observables (genes) to these latent factors. One key issue is identifiability as many latent causal models can result in the same distribution over the observables, necessitating interventions and also mapping them appropriately to represent interventions in the underlying factors. The authors introduce a sparse (pre-conditioned) encoder to mitigate ``causal spillover'' effect, requiring that each gene that involves perturbations is associated with a separate (single) latent factor.
The model is learned with multiple losses, including encoder-decoder reconstruction loss, interventional prediction, and a causal graph loss. The authors also introduce a pre-processing step where control cells are paired with perturbed cells via optimal transport with the idea that this reduces the extent to which cell to cell variation would confuse perturbation effects. The primary contribution in the paper is formulation / architectural since significant theoretical results are borrowed from prior work.

**Strengths:**

The paper is quite well-written and includes a good discussion on challenges with causal representation learning, including intervention spillover and ``mean collapse'' resulting from random pairing of control cells with perturbed ones. The proposed approach seems clear, well-motivated albeit straightforward without significant technical innovations.

**Weaknesses:**

The verbose discussion of effects takes away from technical clarify (steps missing). E.g., the decoder itself or how the effects of perturbations are predicted are not explicated. The underlying causal graph appears to be linear, resolved with the help of DAGMA loss during training. Each intervention in the data has to be associated with a different latent factor. If factors are pathways, this does not hold in practice. Is there a way to mitigate?

**Questions:**

Since the encoder maps observables x to exogenous variables z, how are x reconstructions carried out? By mapping x to z, then z to u via the causal graph, and finally from u back to x?

Do interventions follow the same above steps, starting from a (paired) control cell x, mapped to z, causal graph modified due to intervention on the particular u_i, then resolving the remaining u from the graph, followed by a mapping back to x?

How many samples are there per condition? Is MMD calculated across conditions?

---

> ### Author Response · Authors · 2025-11-26
> **Response 1/2**
>
> **Weaknesses**
>
> 1. Thank you for pointing out that the forward pipeline for perturbation prediction was not described with sufficient technical detail. We agree that the current version emphasizes the motivation behind intervention spillover but does not clearly articulate the computational steps of how a perturbation is applied and propagated through the model.
> As described in Section 3.1, the encoder is preconditioned such that each experimentally targeted gene maps exclusively to a designated meta-pathway. These learned meta-pathways are distinct from curated biological pathways; their purpose is to guarantee the atomic-intervention requirement in latent space. Our model utilizes a $\beta$-VAE framework to regularize the latent space. The process begins by encoding control samples into a latent representation, $\textbf{z}_{\text{obs}}$.
> To model an intervention, we explicitly modify the latent variable associated with the targeted meta-pathway. This modification is performed by a latent modulator, which predicts the change in the latent embedding (i.e., the shift applied to the corresponding $z_i$, see Figure 2). The altered latent representation is then passed through the DAGMA layer, which encodes the learned causal relationships between meta-pathways. DAGMA propagates the intervention throughout the latent space by computing the resulting downstream effects along the directed acyclic graph, producing the updated causal variables $\textbf{u}$. We recognized that placing the detailed schematic (Figure S4) in the appendix made this difficult to confirm. To fix this, we will move Figure S4 and its detailed caption into the main Methods section (Section 3). We will revise the text to explicitly walk the reader through these steps for both reconstruction and counterfactual prediction.
>
> 2. Thank you for raising this critical point regarding the model’s assumption that each intervention must be associated with a distinct latent factor, particularly when these factors are interpreted as biological pathways (BPs). We agree that assuming an intervention targets only a single, existing BP is biologically unrealistic, as genes are highly interconnected and a single knockdown typically impacts multiple BPs simultaneously. However, this concern is mitigated by the fundamental distinction between biologically defined pathways and the latent factors learned by our model. Our framework addresses this by designing the latent factors to be learned meta-pathways. This approach, similar in principle to the architecture in SENA-discrepancy-VAE (de la Fuente et al., 2025), resolves the conflict: the latent factor (causal archetype in SENA-discrepancy-VAE) is modeled as a linear combination of multiple underlying BP activities. This allows one latent factor to act as a causal archetype-center for the entire set of BPs affected by a specific intervention, thereby satisfying the CRL identifiability requirement that each intervention targets a single latent factor. The strict enforcement of this single-node latent intervention is a core design choice in RAPTORGraph, achieved by the preconditioned GraphPathway Encoder which blocks influence to irrelevant factors, thereby eliminating intervention spillover. To clarify this mechanism, we have improved the explanation and labeling in our revised manuscript's Figure 2 and Section 3.1. Finally, the DAGMA loss ensures the relationships between these learned meta-pathways form a valid Directed Acyclic Graph (DAG), learning the system's invariant causal structure. To further enhance the biological interpretation, we plan to perform a Gene Set Enrichment Analysis (GSEA) on the learned factor loadings to quantitatively confirm the biological context of each meta-pathway and provide a clear, testable link between our learned causal archetypes and known biological function.

---

> ### Author Response · Authors · 2025-11-26
> **Response 2/2**
>
> **Questions**
>
> 1. We thank the reviewer for this insightful question regarding the model’s information flow. You are correct: the reconstruction process involves mapping observables to exogenous variables, which are subsequently transformed by the causal graph into endogenous variables before being decoded.
> As detailed in Appendix E.1 and illustrated in Figure S4, the specific reconstruction pathway is as follows:
> **Encoding** ($\textbf{x}$ to $\textbf{z}$): The GraphPathway Encoder maps the observed gene expression $\textbf{x}$ to the exogenous representation $\textbf{z}$.
> **Causal Structuring** ($\textbf{z}$ to $\textbf{u}$): These exogenous variables are passed to the DAG module (DAGMA). This layer effectively solves the structural equations (transforming $\textbf{z}$ via the learned adjacency matrix) to yield the endogenous causal variables.
> **Decoding** ($\textbf{u}$ to $\textbf{x}$): The endogenous variables $\textbf{u}$ are mapped back to the observation space via the decoder to generate the reconstruction.
> Crucially, this architecture supports our soft intervention strategy. For perturbed samples, we encode the intervention as specific changes within the exogenous latent space $\textbf{z}$. These perturbed latents are then processed by the same causal graph and decoder used for control samples. This ensures that while the activity of specific nodes changes, the underlying causal structure remains invariant.
>
> 2. Yes, the intervention workflow follows the same computational sequence as the reconstruction process (detailed in our explanation to your first question), with a specific distinction regarding how the perturbation is applied. To model an intervention, we encode a perturbation label using a latent modulator, which predicts the change in the latent embedding (i.e., the shift applied to the corresponding $z_i$, see Figure 2). The altered latent representation $z_{\text{int}}$ is then passed through the DAGMA layer, which encodes the learned causal relationships between meta-pathways. DAGMA propagates the intervention throughout the latent space by computing the resulting downstream effects along the directed acyclic graph, producing the updated causal variables $\textbf{u}$.
>
> 3. As detailed in Appendix E.1 and Figure S4, we do not modify the causal graph structure itself (i.e., we do not perform hard interventions that remove edges). Instead, we implement soft interventions by modifying the exogenous latent variables $\textbf{z}$ directly.
>
> 4. The dataset contains approximately 108,000 cells across 284 distinct conditions (including controls, single, and double perturbations). Regarding the metric, the Mean Maximum Discrepancy (MMD) is not calculated across different conditions. It is calculated strictly between each model’s predicted distribution for a specific perturbation and the ground truth experimentally observed distribution for that same perturbation. This ensures the metric evaluates the model’s ability to faithfully generate the correct distributional shift for a specific perturbation. Also, we are actively working on including an additional dataset to further demonstrate the generalizability and robustness of RAPTORGraph.

---

> ### Author Response · Authors · 2025-12-03
> **Final Response to Reviewer gVst**
>
> Dear Reviewer gVst,
>
> We sincerely thank you for your positive assessment of our paper, specifically highlighting the "clear, well-motivated" approach and the effective discussion on challenges like intervention spillover. We appreciate the insightful questions regarding the technical implementation.
>
> To help clarify the scope of our experimental effort, we begin with a short summary of the additional analyses performed in response to your concerns:
>
> *   **(W1/Q1/Q2):** We have revised the manuscript to include the full architectural schematic (**Figure 2**) in the main text (**Section 3**), explicitly detailing the $\textbf{x} \rightarrow \textbf{z} \rightarrow \textbf{u} \rightarrow \hat{\textbf{x}}$ reconstruction and intervention paths.
> *   **(W2):** We have clarified the dual nature of our latent factors: rigid "atomic" intervention targets (to satisfy identifiability theory) but flexible "meta-pathway" compositions (to capture biological complexity via dense $W_{12}$/$W_{22}$ connections) (revised **Section 3.1**).
> *   **(Q3):** Clarified metric calculations, confirming MMD is computed per-condition (**Section 4.2**).
>
> Below, we respond to each comment in detail.
>
> ### 1. W1/Q1/Q2: Technical Details of Decoder and Prediction Path
>
> We thank the reviewer for this valuable feedback. You correctly inferred that the reconstruction process involves mapping the observable variables to exogenous variables, which are subsequently transformed by the causal graph into endogenous variables before being decoded.
>
> *   **Clarification:** As detailed in the revised **Section 3** (formerly Appendix E.1), the data flow is:
>     *   **Encoder:** $\textbf{x} \rightarrow \textbf{z}$ (GraphPathway Encoder maps to exogenous representation).
>     *   **Causal Structuring:** $\textbf{z} \rightarrow \textbf{u}$ (DAGMA module solves the structural equations via learned adjacency matrix).
>     *   **Decoder:** $\textbf{u} \rightarrow \hat{\textbf{x}}$ (Dense decoder reconstructs observation).
> *   **Interventions:** Interventions follow the same path, with the addition of the **Latent Modulator**. This module takes the perturbation label and applies a specific shift to the targeted $z_i$, producing $z_{int}$. This modified latent is then passed through the DAGMA layer to resolve downstream effects on other causal variables $\textbf{u}$, which are then decoded.
> *   **Manuscript Revision:** We recognized that placing the detailed schematic (**Figure S4**) in the appendix made this difficult to confirm. To fix this, **we have revised Figure 2 and its detailed caption into the main Methods section (Section 3)**. We have also revised the text to explicitly walk the reader through these steps.
>
> ### 2. W2: Rigidity of Intervention Targets ("One Factor per Intervention")
>
> Thank you for raising this critical point. We agree that assuming an intervention targets only a single biological pathway is biologically simplistic. As such, a causal latent factor could represent the set of biological processes associated with that intervention. This is a **theoretical necessity** to satisfy the 'Atomic Intervention' requirement (Requirement 2 in **Section 2.3**), which is a prerequisite for identifiability in the frameworks we build upon (Zhang et al., 2023; von Kügelgen et al., 2023).
>
> Specifically, our model mitigates this theoretical constraint via the concept of **Learned meta-pathways**:
>
> *   **Atomic Target:** The intervention is architecturally constrained to target a single latent variable $z_i$ (the "anchor"). This prevents the "spillover" problem where a dense encoder would spread the intervention signal across all factors.
> *   **Flexible Composition:** Crucially, while the **intervention** is atomic, the **content** of that latent variable is not. It aggregates information from all other unperturbed genes via the dense $W_{12}$ and $W_{22}$ matrices. This allows a single latent factor to act as a causal archetype for a complex, multi-gene process, thereby satisfying the mathematical necessity of single-node interventions without imposing biologically unrealistic simplicity on the pathway's internal structure. We have revised **Section 3.1**.
>
> ### 3. Q3: Sample Sizes and MMD Calculation
>
> *   **Sample Sizes:** The dataset contains approximately 108,000 cells across 284 distinct conditions (single and double perturbations). The number of cells per condition varies (typically 100-500 cells), reflecting the reality of pooled screens.
> *   **MMD Calculation:** The MK-MMD is **not** calculated across different conditions. It is calculated strictly **per perturbation condition**. For a held-out double perturbation (e.g., GeneA+GeneB), we compute the MK-MMD between the distribution of **predicted** cells and the distribution of **real** cells for that specific condition. The final metric reported in **Table 1** is the average of these condition-specific MK-MMD scores. We have added a sentence to **Section 4.2** to clarify this averaging process.

---

### Author Response · Authors · 2025-11-20
**Global Response**

We would like to thank the reviewers for their comments and suggestions. We believe they have accurately identified the strengths of our work, as well as areas for improvement. The reviewers were generally positive regarding the methodological contributions of RAPTORGraph. For example, reviewers gVst and 8rUJ commended the model’s ability to **“mitigate intervention spillover”**, a limitation that current encoders fail to address. The reviewers also praised RAPTORGraph’s ability to **“perform reverse perturbation analysis”**, predicting causal genes from observed phenotypes, and the **“clear interpretability benefits over the baseline models”** that were demonstrated by the Hit Rate@K results. The reviewers also appreciated the **“solid empirical evaluation”**. Overall, we believe the reviewers agreed that the proposed model has a significant practical impact and **“superior performance”** compared to baseline models, as well as **“architectural novelty”**.

The reviewers raised some weaknesses in the work, such that the **“experimental validation is limited to one dataset”**, raising concerns about its generalizability across different cell types or biological contexts. We also agree with the reviewers that there is a need for **“further biological validation”**, a clearer **“explanation of the model architecture”**, and **“additional ablations”**, specifically to demonstrate the contribution of Optimal Transport (OT).

We believe these concerns can be addressed in the current review process. Due to time constraints and to facilitate discussion with the reviewers, we have compiled a list of improvements and contributions that the reviewers have proposed and that we have already implemented. We plan to address the remaining questions and concerns, and upload an updated manuscript in the coming days.

1. The captions of Tables 1–3 have been updated to clearly state that the values represent the mean and variance across three experimental runs (using different random seeds).
2. We have clarified the calculation of the MK-MMD metric in Section 4.2, explaining that it is computed per-condition and subsequently averaged.
3. We have added a clarifying sentence to the discussion on intervention spillover, distinguishing between the architectural spillover that our model prevents and the real biological correlations that it is designed to model.
4. A discussion regarding the theoretical robustness of OT, particularly in handling less structured datasets, has been added to Appendix E.4.
5. We have significantly updated Section 3.1 and Figure 2 to provide a clearer explanation of our preconditioning method and its architectural components.

As mentioned above, we plan to upload the updated version of the manuscript, which incorporates the remaining concerns raised by the reviewers and requires extra working hours:

1. We are incorporating a second dataset (Replogle et al., 2020, Nature Biotechnology) to strengthen the generalizability of our findings.
2. We are running ablations that use random pairing rather than OT alignment to quantify the impact of OT.
3. We will benchmark the computational overhead introduced by the OT alignment step to provide a complete picture of its cost-benefit trade-off.
4. We are incorporating additional assessments for biological discovery (e.g., based on Gene Set Enrichment Analysis, among others) to further validate the biological interpretability of our model’s causal latent factors, and how they are linked to known biological processes.

---

### Author Response · Authors · 2025-12-03
**Final Global Response to Area Chair**

We understand the unique circumstances regarding the review process and thank the area chair for taking the additional time dedicated to evaluating our work. In the following, we would like to summarize our rebuttal to help the area chair make an informed decision. We are confident that the extensive revisions and new experiments conducted during the rebuttal period directly address all concerns raised by the Reviewers.

Firstly, the reviewers were uniformly positive regarding the methodological contributions of RAPTORGraph, such as:

*   **Methodological Contribution:** Reviewers **gVst** and **8rUJ** praised that our work raised, for the first time, the intervention spillover problem in CRL. They specifically commended our model’s ability to "mitigate it", acknowledging this as a critical limitation that current encoders fail to address.
*   **Interpretability:** All reviewers further praised RAPTORGraph’s "impressive" ability to "perform reverse perturbation analysis", predicting causal genes from observed phenotypes. They noted the "clear interpretability benefits over the baseline models" demonstrated by our Hit Rate@K results.
*   **Performance:** The reviewers appreciated the "solid empirical evaluation", agreeing that the proposed model has a significant practical impact and demonstrates "superior performance" compared to baseline models, alongside "architectural novelty".

While we are encouraged by the reviewers' recognition of our methodological contributions, we have significantly improved our paper by addressing all their feedback and questions. To this end, **we have executed a comprehensive suite of new experiments and analyses that directly resolve these concerns**. Furthermore, **we have updated the manuscript to improve clarity, moving critical technical details** (such as the full reconstruction path and DAGMA optimization strategy) to the main text. The following tables map each concern to the specific new contributions resolving it.

### 1. Data Diversity & Generalization

| Concern Addressed | Reviewers | New Experiment / Content |
| :--- | :--- | :--- |
| Limited dataset diversity and justification for a single-dataset. | 6bPr-W1, 8rUJ-W1, 6bPr-Q3 | **New Dataset Benchmark:** We added the Replogle et al. (2020) dataset (**Table S6**). We have also included a rationale for the importance of the Norman-CPA dataset (**Sec. 5**). |

### 2. Model Architecture & Ablations

| Concern Addressed | Reviewers | New Experiment / Content |
| :--- | :--- | :--- |
| Ablation studies needed to assess: 1) Contribution of OT (mitigate the effect of Mean Collapse), and 2) Contribution of encoder preconditioning (assess the rigidity of the encoder). | 6bPr-W2, 8rUJ-W2, 8rUJ-W4, 8rUJ-Q4 | **New ablation studies** showcasing the value of OT (**Sec. F4, F5**), and the preconditioning encoder in RAPTORGraph. Note that the baseline dVAE underperforms in predicting non-additive interactions (**Table 2**) validating the necessity of preconditioning. |
| Further assessment of hyperparameter sensitivity and parameter Trade-offs needed. | 6bPr-Q2, 8rUJ-Q1 | **New sensitivity analysis** of model’s hyperparameters at the requested modules: $\beta$-VAE and $\delta_{DAGMA}$ (**Sec F.6**) |
| Assessment of the OT’s computational overhead | 6bPr-W4 | **New training time Benchmark:**OT overhead (**Sec F.4**) |

### 3. Biological Validation & Interpretability

| Concern Addressed | Reviewers | New Experiment / Content |
| :--- | :--- | :--- |
| Lack of Biological Validation and potential hidden (biological) confounders in the DAG | 6bPr-W3, 8rUJ-W3, 8rUJ-Q3 | **A new in-depth biological validation** of causal latent factors is presented via Gene Set Enrichment Analysis (**Sec 5, F.2**). We have further elaborated on benefits of the preconditioning module to reduce potential confounders in the DAG (**Sec E.2.1**). |

### 4. Technical Clarifications

| Concern Addressed | Reviewers | New Experiment / Content |
| :--- | :--- | :--- |
| Lack of clarity on the decoder and prediction of unseen perturbations | gVst-W1, gVst-Q1, gVst-Q2 | **Revised Schematic:** **Fig 2, Sec 5.** |
| Lack of clarity on the rigidity of preconditioning and its identifiability guarantees | gVst-W2, 6bPr-Q1, 8rUJ-Q2 | **Revised Text:** **Sec 2.4, Sec 3.1** (Atomic Interventions) |
| DAGMA vs. Reconstruction Trade-off | 8rUJ-Q1 | **Revised Text:** **Sec 3.2** (Gradient Detachment) |
| OT Generalization | 8rUJ-Q4 | **Discussion:** OT Robustness to Confounders (**Sec E.4**) |
| Significance of Results | 6bPr-Q4 | **Clarification:** Statistical Significance (**Table 2**) |
| Metric Details | gVst-Q3 | **Clarification:** MMD Calculation (**Sec 4.2**) |
| General Presentation | All | **Manuscript Revision:** Clarity Improvements |

We hope this summary helps the area chair navigate our revised manuscript and assess how well our rebuttal addresses the reviewers' concerns. Individual responses to the reviewers have also been posted below.

---

### Meta-Review · Area_Chair_oaQS · 2026-01-05

**Summary:**

This paper proposes a causal representation learning framework, RAPTORGraph, for modeling single-cell perturbation responses using a sparse encoder to mitigate intervention spillover, a DAGMA-based causal graph, and optimal transport pre-processing to address mean collapse.

**Reviewer Concerns:**

Although the paper has some merits, such as clear problem formulation and a comprehensive evaluation on a standard dataset, the issues raised by the reviews are critical. For instance, the lack of identifiability guarantees and model's rigidity and unrealistic assumption (Reviewer gVst), the limited evaluation to a single dataset and lack of biological validation for the learned causal graph (Reviewer 6bPr), and the potential bias introduced by the constrained encoder and unclear biological interpretability of the discovered DAG (Reviewer 8rUJ). Although the authors address some issues in responses, the paper still needs a major revision before it can be accepted.

**Reviewer Scores:**

gVst and 6bPr: Would not change score; concerns about identifiability and technical clarity are fundamental.

8rUJ: Score might remain the same; while appreciating the motivation, the need for broader validation and proof of biological interpretability to support identifiability claims persists.

---

### Decision · Program_Chairs · 2026-01-26

Reject